# Policy Gradient for Rectangular Robust Markov Decision Processes

**Navdeep Kumar**[*]
Technion

**Esther Derman**
MILA, Université de Montréal

**Matthieu Geist**
Goodle Deepmind

**Kfir Levy**
Technion

**Shie Mannor**
Technion, NVIDIA Research

## Abstract

Policy gradient methods have become a standard for training reinforcement learning agents in a scalable and efficient manner. However, they do not account for transition uncertainty, whereas learning robust policies can be computationally expensive. In this paper, we introduce robust policy gradient (RPG), a policy-based method that efficiently solves rectangular robust Markov decision processes (MDPs). We provide a closed-form expression for the worst occupation measure. Incidentally, we find that the worst kernel is a rank-one perturbation of the nominal. Combining the worst occupation measure with a robust Q-value estimation yields an explicit form of the robust gradient. Our resulting RPG can be estimated from data with the same time complexity as its non-robust equivalent. Hence, it relieves the computational burden of convex optimization problems required for training robust policies by current policy gradient approaches.

## 1 Introduction

Markov decision processes (MDPs) provide an analytical framework to solve sequential decision-making problems and seek the best performance in a fixed environment. Since the resulting policy can be highly sensitive to parameter values [18], the robust MDP setting alternatively maximizes return under the worst scenario, thus yielding robustness to uncertain environments [20, 12]. In practice, the robust MDP paradigm quantifies the level of uncertainty through a set $\mathcal{U}$ determining the possible range of model perturbations. Then, a policy is said to be robust-optimal if it reaches maximal performance under the most adversarial model within the uncertainty set. Developing efficient solvers for robust MDPs is of great interest, as it can lead to behavior policies with generalization guarantees [34].

If not computationally expensive, robust MDPs can be strongly NP-hard [33]. Thus, to preserve tractability, we commonly assume that $\mathcal{U}$ is convex and $s$-rectangular, i.e., $\mathcal{U} = \times_{s \in \mathcal{S}} \mathcal{U}_s$ [20, 12, 33]. Well established in the robust reinforcement learning (RL) literature [8, 3, 9, 10, 28], the latter assumption means that the overall uncertainty should be designed independently for each state. Further simplification may consider $(s, a)$-rectangular uncertainty sets of the form $\mathcal{U} = \times_{(s,a) \in \mathcal{X}} \mathcal{U}_{(s,a)}$, albeit this naturally leads to more conservative strategies. In any case, planning in robust MDPs can be computationally costly, as it involves successive max-min problems [9, 1, 33]. To address this issue, the works [3, 14] have established an equivalence between robustness and regularization in RL in order to derive efficient robust planning methods for $s$ and $(s, a)$-rectangular robust MDPs. Indeed, it appears that resorting to proper regularization instead of solving a minimization problem can yield robust behavior without requiring the polynomial time complexity of convex optimization problems [3].

---

[*]Contact author `navdeepkumar@campus.technion.ac.il`

37th Conference on Neural Information Processing Systems (NeurIPS 2023).

Alternatively to planning, policy gradient algorithms (PG) directly learn an optimal policy by applying gradient steps towards better performance [25]. Due to its scalability, ease of implementation, and adaptability to many different settings such as model-free and continuous state-action spaces [13, 24], PG has become the workhorse of RL. Although regularization techniques such as max-entropy [7] or Tsallis [15] have shown robust behavior without impairing computational cost, they only account for adversarial reward [2, 6, 3]. Differently, robust PG formulations (RPG) formulations aim to address uncertainty to reward *and* transition functions.

Despite their ability to propel robust behavior, RPG methods that target robust optimal policies are still rare in the RL literature. The global convergence of RPG established in [16, 30] already ensures global convergence of our proposed algorithm, but motivates a practical method for estimating the gradients. Indeed, [16, 30] occult the estimation part, as they assume full access to the policy gradient. Alternatively, the inner loop solution proposed in [30][Sec. 4.1] requires solving convex optimization problems to find the worst model, which represents a larger time complexity of $O(S^4 A \log \epsilon^{-1})$ for $(s, a)$-rectangular, or $O(S^4 A^3 \log \epsilon^{-1})$ for $s$-rectangular uncertainty sets. These worst kernel and reward models are needed to compute RPG using the policy gradient theorem [26]. Other approaches that elicit an expression for RPG rely on a specific type of uncertainty set such as reward uncertainty with known kernel [3, 5], $r$-contaminated kernel with known reward [32], or $(s, a)$-rectangular uncertainty [16], whereas we aim to tackle more general robust MDPs.

In this work, we introduce an RPG method for both $s$ and $(s, a)$-rectangular ball-constrained uncertainty sets, with similar complexity as non-robust PG. Our approach provides a closed-form expression of RPG without relying on an oracle while applying to the most common robust MDPs. To this end, we derive the worst reward and transition functions, thus revealing the adversarial nature of the corresponding uncertainty set. Surprisingly, we also find that the worst kernel is a rank-one perturbation of the nominal kernel. Leveraging this rank-one perturbation enables us to derive a robust occupation measure. We concurrently propose an alternative definition of the robust Q-value together with an efficient way to estimate it. Combining these results enables us to obtain RPG in closed form. Our resulting RPG update requires $O(S^2 A \log \epsilon^{-1})$ computations, thus showing similar time complexity as non-robust PG.

To summarize our contributions: *(i)* We establish the worst reward and transition models in closed-form; *(ii)* We show that the worst-case transition function is a rank-one perturbation of the nominal; *(iii)* We introduce alternative robust Q-values that can be evaluated through efficient Bellman recursion while retrieving the robust value function; *(iv)* We establish an expression of RPG that can be estimated with similar time complexity as non-robust PG. Experiments show that our RPG speeds up state-of-the-art robust PG updates by 2 orders of magnitude.

## 2 Related work

Although some previous works use gradient methods to learn robust policies, they seek empirical robustness to adversarial behavior rather than robust MDP solutions [21, 29, 4]. In that sense, our study differs from adversarial RL as we explicitly optimize the max-min objective to find a robust optimal policy. Accordingly, the risk-averse approach focuses on the *internal uncertainty* due to the stochasticity of the system, whereas robust RL addresses the *external uncertainty* of the system's dynamics. As a result, common risk-averse objectives can be reformulated as robust problems with specific uncertainty sets [27].

Previous studies that did aim to derive robust policy-based methods are [3, 32, 30]. These are summarized in Table 1, which also displays the complexity of existing approaches. [3] established RPG for $s$-rectangular reward-robust MDPs, i.e., robust MDPs with uncertain reward but given kernel. Although it applies to general norms, their result does not account for transition perturbation. Differently, in [32], the authors introduced RPG for $r$-contaminated MDPs, i.e., robust MDPs with uncertainty set $\mathcal{U} := \{R_0\} \times [(1-r)P_0 + r\Delta_{\mathcal{S}}^{\mathcal{S} \times \mathcal{A}}]$. Although it has similar complexity as non-robust PG, by construction, their setting is limited to $(s, a)$-rectangularity with known reward and mixed transition. As such, the proof techniques in [32] are tailor-made to the $r$-contamination framework and do not apply to more general robust MDPs. In fact, we remark that the $r$-contamination setting is equivalent to the action robustness approach introduced in [29], which emphasizes its limitation to action perturbation. Differently, our RPG holds whenever the worst kernel is a rank-one perturbation of the nominal transition function (see Lemma 4.4).

The work [16] provides a convergence proof of robust policy mirror-descent in the $(s, a)$-rectangular case, whereas we study robust policy optimization for $s$-rectangular uncertainty sets. In fact, its restriction to the $(s, a)$-case prevents us from transposing the analysis to our setting. This is due to the fact that the standard robust Bellman operator on $Q$-functions can no longer be applied on $s$-rectangular sets. To address generic robust MDPs, [30] recently introduced RPG for general uncertainty sets. Their gradient update has a complexity of $O(S^6 A^4 \epsilon^{-4})$, which is more expensive than non-robust PG by a factor of $S^4 A^3 \epsilon^{-4}$. Both works [16, 30] additionally assume access to an oracle gradient of the robust return with respect to the transition model. Avoiding this oracle assumption naturally leads to even higher time complexity in [30] which is not scalable. At the same time, the two works [16, 30] guarantee global convergence of projected robust gradient iterates, thus establishing the potential promise of RPG. In fact, equipped with RPG convergence, the remaining challenge in making it practical is to efficiently estimate the gradient. This represents the main focus of our study: We aim to explicit an RPG method that generalizes existing results on specific uncertainty sets [3, 32] while holding for $s$-rectangular robust MDPs.

Table 1: Time complexity of RPG update according to the type of uncertainty set. For conciseness, the displayed complexity hides logarithmic factors in $A$ and $S$. Our RPG method has the same complexity as non-robust PG while it generalizes other RPG methods with similar efficiency.

| UNCERTAINTY SET $\mathcal{U}$ | TIME COMPLEXITY | REFERENCE |
| --- | --- | --- |
| $\{R_0\} \times \{P_0\}$ | $S^2 A \log \epsilon^{-1}$ | [26] |
| $\{R_0\} \times [(1 - r)P_0 + r\Delta_\mathcal{S}^{\mathcal{S} \times \mathcal{A}}]$ | $S^2 A \log \epsilon^{-1}$ | [32] |
| $(s, a)$-rectangular ball $\mathcal{U}_p^{\mathtt{sa}}$ | $S^2 A \log \epsilon^{-1}$ | **This work** |
| $(s, a)$-rectangular, convex $\mathcal{U}^{\mathtt{sa}}$ | $S^4 A \log \epsilon^{-1}$ | Convex optimization |
| $s$-rectangular ball $\mathcal{U}_p^{\mathtt{s}}$ | $S^2 A \log \epsilon^{-1}$ | **This work** |
| $s$-rectangular ball $(R_0 + \mathcal{R}_p^{\mathtt{s}}) \times \{P_0\}$ | $S^2 A \log \epsilon^{-1}$ | [3] |
| $s$-rectangular, convex $\mathcal{U}^{\mathtt{s}}$ | $S^4 A^3 \log \epsilon^{-1}$ | Convex optimization |
| $s$-rectangular, convex $\mathcal{U}^{\mathtt{s}}$ | $S^6 A^4 \epsilon^{-4}$ | [30] |
| $s$-rectangular, non-convex $\mathcal{U}^{\mathtt{s}}$ | NP-hard | [33] |
| Non-rectangular, convex $\mathcal{U}$ | NP-hard | [33] |

# 3 Preliminaries

**Notation:** We denote the cardinal of an arbitrary finite set $\mathcal{Z}$ by $|\mathcal{Z}|$. Given two real functions $\mathbf{a}, \mathbf{b} : \mathcal{Z} \to \mathbb{R}$, their inner product is $\langle \mathbf{a}, \mathbf{b} \rangle_{\mathcal{Z}} := \sum_{z \in \mathcal{Z}} \mathbf{a}(z) \mathbf{b}(z)$, which induces the $\ell_2$-norm $\|\mathbf{a}\|_2 := \sqrt{\langle \mathbf{a}, \mathbf{a} \rangle_{\mathcal{Z}}}$. More generally, the $\ell_p$-norm of $\mathbf{a}$ is denoted by $\|\mathbf{a}\|_p$ whose conjugate norm is $\|\mathbf{a}\|_q := \max_{\|\mathbf{b}\|_p \leq 1} \langle \mathbf{a}, \mathbf{b} \rangle_{\mathcal{Z}}$ with $q^{-1} = 1 - p^{-1}$, by Hölder's inequality. The vector of all zeros (resp. all ones) with appropriate dimensions is denoted by $\mathbf{0}$ (resp. $\mathbf{1}$); the probability simplex over $\mathcal{Z}$ by $\Delta_{\mathcal{Z}} := \{\mathbf{a} : \mathcal{Z} \to \mathbb{R}_+ \,|\, \langle \mathbf{a}, \mathbf{1} \rangle_{\mathcal{Z}} = 1\}$, and $I_{\mathcal{Z}}$ designates the identity matrix in $\mathbb{R}^{\mathcal{Z} \times \mathcal{Z}}$. Given $v \in \mathbb{R}^{\mathcal{Z}}$, we finally define the variance function as $\kappa_q(v) = \min_{w \in \mathbb{R}} \|v - w\mathbf{1}\|_q$ and the mean function as $\omega_q(v) \in \arg\min_{w \in \mathbb{R}} \|v - w\mathbf{1}\|_q$ (see Tab. 2 for their closed-form expression when $q \in \{1, 2, \infty\}$).

## 3.1 Markov Decision Processes

A Markov decision process (MDP) is a tuple $(\mathcal{S}, \mathcal{A}, \gamma, \mu, P, R)$ such that $\mathcal{S}$ and $\mathcal{A}$ are finite state and action spaces of cardinal $S$ and $A$ respectively, $\gamma \in [0, 1)$ is a discount factor and $\mu \in \Delta_{\mathcal{S}}$ the initial state distribution. Denoting $\mathcal{X} := \mathcal{S} \times \mathcal{A}$, the couple $(P, R)$ corresponds to the MDP model with $P : \mathcal{X} \to \Delta_{\mathcal{S}}$ being a transition kernel and $R : \mathcal{X} \to \mathbb{R}$ a reward function. A policy $\pi : \mathcal{S} \to \Delta_{\mathcal{A}}$ maps each state to a probability distribution over $\mathcal{A}$, and we denote by $\Pi$ the set of such functions. For any policy $\pi \in \Pi$, $R^\pi \in \mathbb{R}^{\mathcal{S}}$ is the expected immediate reward defined as $R^\pi(s) := \langle \pi_s, R(s, \cdot) \rangle_{\mathcal{A}}, \quad \forall s \in \mathcal{S}$, where $\pi_s$ is a shorthand for $\pi(\cdot|s)$. We similarly define the stochastic matrix induced by $\pi$ as $P^\pi(s'|s) := \langle \pi_s, P(s'|s, \cdot) \rangle_{\mathcal{A}}, \quad \forall s, s' \in \mathcal{S}$, and extend the occupation measure to an arbitrary initial vector $\nu \in \mathbf{R}^{\mathcal{S}}$ by defining

$$d_{P,\nu}^\pi := \nu^\top (I_{\mathcal{S}} - \gamma P^\pi)^{-1}.$$

Note that the initial vector here is not necessarily a probability measure: it can be the initial state distribution, but also the balanced value function introduced in Sec. 4[Eq. (2)]. The performance measure we aim to maximize is the value function $v^\pi_{(P,R)} := (I_\mathcal{S} - \gamma P^\pi)^{-1} R^\pi$, or alternatively, the return $\rho^\pi_{(P,R)} := \langle \mu, v^\pi_{(P,R)} \rangle_\mathcal{S}$. We denote the optimal value function (resp. optimal return) by $v^*_{(P,R)} = \max_{\pi \in \Pi} v^\pi_{(P,R)}$ (resp. $\rho^*_{(P,R)} = \langle \mu, v^*_{(P,R)} \rangle$). It can be obtained using Bellman operators, which are defined as $T^\pi_{(P,R)} v := R^\pi + \gamma P^\pi v$ and $T^*_{(P,R)} v := \max_{\pi \in \Pi} T^\pi_{(P,R)} v, \quad \forall v \in \mathbb{R}^\mathcal{S}$, respectively [22]. For any vector $v \in \mathbb{R}^\mathcal{S}$, we associate its Q-function $Q \in \mathbb{R}^\mathcal{X}$ such that

$$Q(s,a) = R(s,a) + \gamma \langle P(\cdot|s,a), v \rangle_\mathcal{S}, \quad \forall (s,a) \in \mathcal{X}.$$

Table 2: Expressions of the $q$-mean, the $q$-variance, and its gradient. We assume that the vector $v$ is sorted, i.e., $v(s_i) \geq v(s_{i+1}), \forall i \in \{1, 2, \cdots, S\}$, and denote $n_l := \lfloor (S+1)/2 \rfloor, n_u := \lceil (S+1)/2 \rceil$.

| | $\omega_q(v)$ | $\kappa_q(v)$ | $\nabla_v \kappa_q(v)$ |
|---|---|---|---|
| $q$ | $\arg\min_{w \in \mathbb{R}} \|v - w\mathbf{1}\|_q$ | $\min_{\omega \in \mathbb{R}} \|v - \omega\mathbf{1}\|_q$ | $\frac{\partial \kappa_q(v)}{\partial v(s_i)}$ |
| $\infty$ | $\frac{v(s_1) + v(s_S)}{2}$ | $\frac{v(s_1) - v(s_S)}{2}$ | $\begin{cases} \frac{1}{2} & \text{if } i = 1 \\ -\frac{1}{2} & \text{if } i = S \\ 0 & \text{o.w.} \end{cases}$ |
| $2$ | $\frac{\sum_{i=1}^S v(s_i)}{S}$ | $\sqrt{\sum_{i=1}^S (v(s_i) - \omega_2(v))^2}$ | $\frac{v(s_i) - \omega_2(v)}{\kappa_2(v)}$ |
| $1$ | $\frac{v(s_{n_l}) + v(s_{n_u})}{2}$ | $\sum_{i=1}^{n_l} (v(s_i) - v(s_{S-i}))$ | $\begin{cases} 1 & \text{if } i < n_l \\ -1 & \text{if } i > n_u \\ 0 & \text{o.w.} \end{cases}$ |

With a slight abuse of notation, we can similarly define a Bellman operator over Q-values as

$$T^\pi_{(P,R)} Q(s,a) := R(s,a) + \gamma \sum_{(s',a') \in \mathcal{X}} P(s'|s,a) \pi_{s'}(a') Q(s',a'), \quad \forall (s,a) \in \mathcal{X}.$$

## 3.2 Robust Markov Decision Processes

In a robust MDP setting, we assume that $(P, R) \in \mathcal{U}$ and aim to maximize return under the worst model from the set. We denote the robust performance of a policy $\pi \in \Pi$ by $\rho^\pi_\mathcal{U} := \min_{(P,R) \in \mathcal{U}} \rho^\pi_{(P,R)}$. It is maximal when it reaches $\rho^*_\mathcal{U} := \max_{\pi \in \Pi} \rho^\pi_\mathcal{U}$ at an optimal robust policy $\pi^*_\mathcal{U} \in \arg\max_{\pi \in \Pi} \rho^\pi_\mathcal{U}$. When considering the robust value function $v^\pi_\mathcal{U} := \min_{(P,R) \in \mathcal{U}} v^\pi_{(P,R)}$, we further need to assume that $\mathcal{U}$ is convex and rectangular so that an optimal robust policy realizing $v^*_\mathcal{U} := \max_\pi v^\pi_\mathcal{U}$ can be computed in polynomial time [33]. We thus assume $\mathcal{U}$ to be convex and rectangular in the remainder of this work. Specifically, we denote an $(s,a)$-rectangular uncertainty set by $\mathcal{U}^{\mathsf{sa}} := \times_{(s,a) \in \mathcal{X}} (\mathcal{P}_{(s,a)}, \mathcal{R}_{(s,a)})$. It represents a particular case of $s$-rectangular uncertainty which we similarly denote by $\mathcal{U}^\mathsf{s} := \times_{s \in \mathcal{S}} (\mathcal{P}_s, \mathcal{R}_s)$. In both cases, there exists an optimal robust policy that is stationary, although all optimal ones may be stochastic [33].

Similarly to non-robust MDPs, robust MDPs can be solved through Bellman recursion. Indeed, the robust value function $v^\pi_\mathcal{U}$ (resp., optimal robust value function $v^*_\mathcal{U}$) is known to be the unique fixed point of the robust Bellman operator $T^\pi_\mathcal{U} v := \min_{(P,R) \in \mathcal{U}} T^\pi_{(P,R)} v$ (resp., the optimal robust Bellman operator $T^*_\mathcal{U} v := \max_{\pi \in \Pi} T^\pi_\mathcal{U} v$), $\forall v \in \mathbb{R}^\mathcal{S}$, both being $\gamma$-contractions for the sup-norm. Although this ensures linear convergence of robust value iteration, the evaluation of each Bellman operator can still be prohibitive for practical use.

### 3.2.1 Ball Constrained Uncertainty set

To facilitate the computation of robust Bellman updates, we consider uncertainty sets that are centered around a nominal model $(P_0, R_0)$, i.e., of the form $\mathcal{U} = (P_0, R_0) + (\mathcal{P}, \mathcal{R})$, and constrained according

to $\ell_p$-norm balls [3, 14, 9, 1]. In the $(s, a)$-rectangular case, the corresponding uncertainty set is denoted by $\mathcal{U}_p^{\mathsf{sa}} := \mathcal{R}_p^{\mathsf{sa}} \times \mathcal{P}_p^{\mathsf{sa}} = \times_{(s,a) \in \mathcal{X}} (\mathcal{P}_{(s,a)}, \mathcal{R}_{(s,a)})$ where for any $(s, a) \in \mathcal{X}$,

$$\mathcal{R}_{(s,a)} = \{r \in \mathbb{R} \mid |r| \leq \alpha_{s,a}\}, \quad \text{and} \quad \mathcal{P}_{(s,a)} = \{p \in \mathbb{R}^{\mathcal{S}} \mid \langle p, \mathbf{1} \rangle_{\mathcal{S}} = 0, \|p\|_p \leq \beta_{s,a}\}.$$

Similarly, an $s$-rectangular norm-constrained uncertainty is denoted by $\mathcal{U}_p^{\mathsf{s}} := \times_{s \in \mathcal{S}} (\mathcal{P}_s, \mathcal{R}_s)$ where for any $s \in \mathcal{S}$,

$$\mathcal{R}_s = \{r \in \mathbb{R}^{\mathcal{A}} \mid \|r\|_p \leq \alpha_s\}, \quad \text{and} \quad \mathcal{P}_s = \{p \in \mathbb{R}^{\mathcal{X}} \mid \langle p(\cdot, a), \mathbf{1} \rangle_{\mathcal{S}} = 0 \quad \forall a \in \mathcal{A}, \|p\|_p \leq \beta_s\}.$$

In both cases, the kernel uncertainty set conceals linear constraints ensuring all entries in $P_0 + \mathcal{P}$ are non-negative. Indeed, we generally ignore $P_0$ to satisfy these constraints in practice [23]. Although it may include absurd models and unnecessarily lead to conservative policies, this proxy region is appropriate for model-free robust learning. Moreover, the norm-ball structure on uncertainty sets above enables us to compute robust Bellman updates with similar time complexity as non-robust ones using regularization [3, 14]. We formalize this below.

**Proposition 3.1.** *([14, Thm. 2-3].) For any policy $\pi \in \Pi$ and any rectangular $\ell_p$-ball-constraint uncertainty set, the robust Bellman operator is equivalent to its regularized form:*

$$(T_{\mathcal{U}}^{\pi} v)(s) = T_{(P_0, R_0)}^{\pi} v(s) + \Omega_q(\alpha, \beta, v),$$

*where $\Omega_q(\alpha, \beta, v) := -\langle \pi_s, \alpha_{s,\cdot} + \gamma \kappa_q(v) \beta_{s,\cdot}^P \rangle_{\mathcal{A}}$ for $(s, a)$-rectangular uncertainty $\mathcal{U}_p^{\mathsf{sa}}$, and $\Omega_q(\alpha, \beta, v) := -(\alpha_s + \gamma \beta_s \kappa_q(v)) \|\pi_s\|_q$ for $s$-rectangular uncertainty $\mathcal{U}_p^{\mathsf{s}}$.*

In the following, we leverage the regularized formulation of robust value functions to explicitly derive RPG for rectangular $\ell_p$-ball uncertainty sets.

### 3.2.2 Robust Gradient Method

Since the robust return can be non-differentiable, we need to follow the projected sub-gradient ascent rule in order to optimize the robust return, namely, update $\pi_{k+1} := \mathbf{proj}_{\Pi}(\pi_k + \eta \partial_{\pi} \rho_{\mathcal{U}}^{\pi_k})$ where

$$\partial_{\pi} \rho_{\mathcal{U}}^{\pi} := \nabla_{\pi} \rho_{(P,R)}^{\pi} \Big|_{(P,R)=(P_{\mathcal{U}}^{\pi}, R_{\mathcal{U}}^{\pi})}, \tag{1}$$

$\eta$ is the learning rate, $\mathbf{proj}_{\Pi}$ denotes the orthogonal projection on $\Pi$, and $(P_{\mathcal{U}}^{\pi}, R_{\mathcal{U}}^{\pi})$ is the worst model associated with $\pi \in \Pi$ and $\mathcal{U}$, i.e., $(P_{\mathcal{U}}^{\pi}, R_{\mathcal{U}}^{\pi}) \in \arg\inf_{(P,R) \in \mathcal{U}} \rho_{(P,R)}^{\pi}$.

Given oracle access to sub-gradient $\partial \rho_{\mathcal{U}}^{\pi}$, projected gradient ascent converges to an $\epsilon$-optimal policy $\pi_{\mathcal{U}}^*$. Moreover, under similar conditions as in the non-robust setting, projected gradient ascent holds an iteration complexity of $O(S^4 A^2 \epsilon^{-4})$ [30]. Yet, the sub-gradient in (1) is generally intractable, particularly because general convex uncertainty sets may yield NP-hard complexity. Instead, we propose to focus on ball-constrained uncertainty sets in order to efficiently compute RPG updates.

## 4 Towards RPG: Expressing the worst quantities

In this section, we provide all the ingredients needed for deriving RPG. Before diving into the gradient expression, we first settle on the general framework of policy gradient. Secondly, in Sec. 4.1, we focus on expressing the worst model according to the nominal explicitly. Surprisingly, we find that the worst transition kernel is a rank-one perturbation of the nominal. This finding enables us to derive the robust occupancy measure, i.e., the visitation frequency of the worst kernel in Sec. 4.2. As a last piece, in Sec. 4.3, we propose an alternative definition of robust Q-value and show that it can be estimated from a specific Bellman recursion.

Consider again the projected gradient ascent rule:

$$\pi_{k+1} := \mathbf{proj}_{\Pi}(\pi_k + \eta \partial_{\pi} \rho_{\mathcal{U}}^{\pi_k}).$$

By definition of the sub-gradient in (1) and applying the standard PG theorem [26], it holds that:

$$\partial_{\pi} \rho_{\mathcal{U}}^{\pi} = \sum_{(s,a) \in \mathcal{X}} d_{\mathcal{U}}^{\pi}(s) Q_{\mathcal{U}}^{\pi}(s, a) \nabla \pi_s(a),$$

where $Q_{\mathcal{U}}^{\pi} := Q_{(P_{\mathcal{U}}^{\pi}, R_{\mathcal{U}}^{\pi})}^{\pi}$ is the Q-value associated with the worst-case model, and $d_{\mathcal{U}}^{\pi} := d_{P_{\mathcal{U}}^{\pi}}^{\pi}$ the occupation measure of the worst transition kernel. In fact, for the uncertainty sets we focus on in this work, the worst Q-value $Q_{\mathcal{U}}^{\pi}$ retrieves the common definition of robust Q-value [20, 28] (see the appendix for a detailed discussion). Therefore, for conciseness and with a slight abuse, we shall designate $Q_{\mathcal{U}}^{\pi}$ by the robust Q-value, and $d_{\mathcal{U}}^{\pi}$ by the robust visitation frequency. The remaining question is how to compute these quantities and in particular, can we efficiently find the worst parameters $(P_{\mathcal{U}}^{\pi}, R_{\mathcal{U}}^{\pi})$? The following part of our study aims to address these questions.

Given an uncertainty set $\mathcal{U}$, let first define the normalized and balanced robust value function as:

$$u_{\mathcal{U}}^{\pi}(s) := \frac{\text{sign}(v_{\mathcal{U}}^{\pi}(s) - \omega_q(v_{\mathcal{U}}^{\pi}))\|v_{\mathcal{U}}^{\pi}(s) - \omega_q(v_{\mathcal{U}}^{\pi})\|^{q-1}}{\kappa_q(v_{\mathcal{U}}^{\pi})^{q-1}}. \tag{2}$$

By construction, it has zero mean and unit norm, i.e., $\langle u_{\mathcal{U}}^{\pi}, \mathbf{1} \rangle_{\mathcal{S}} = 0$ and $\|u_{\mathcal{U}}^{\pi}\|_p = 1$. In fact, as stated in the result below, $u_{\mathcal{U}}^{\pi}$ is the gradient of the $q$-variance function, and correlates with the (unnormalized, unbalanced) robust value function according to the same $q$-variance.

**Proposition 4.1.** *For any policy $\pi \in \Pi$ and $\ell_p$-ball rectangular uncertainty set, the following holds:*

$$u_{\mathcal{U}}^{\pi} = \nabla_v \kappa_q(v) \Big|_{v=v_{\mathcal{U}}^{\pi}},$$

$$\langle u_{\mathcal{U}}^{\pi}, v_{\mathcal{U}}^{\pi} \rangle = \kappa_q(v_{\mathcal{U}}^{\pi}).$$

### 4.1 Worst Kernel and Reward

In the following results, we explicit the relationship between the nominal and the worst-case model for $(s, a)$ and $s$-rectangular $\ell_p$-balls. We will then leverage this relationship to compute the robust Q-values and the robust occupation measure, both necessary for RPG.

**Theorem 4.2** ($(s, a)$-rectangular case). *Given uncertainty set $\mathcal{U} = \mathcal{U}_p^{sa}$ and any policy $\pi \in \Pi$, the worst model is related to the nominal one through:*

$$R_{\mathcal{U}}^{\pi}(s, a) = R_0(s, a) - \alpha_{s,a} \qquad and \qquad P_{\mathcal{U}}^{\pi}(\cdot|s, a) = P_0(\cdot|s, a) - \beta_{s,a} u_{\mathcal{U}}^{\pi}.$$

Based on Thm. 4.2, it follows that in the $(s, a)$-rectangular case, the worst reward function is independent of the employed policy. As we establish in Thm. 4.3 below, this no longer applies under $s$-rectangularity. In either case, the worst kernel is policy-dependent, discouraging the system to move toward high-rewarding states and directing it to low-rewarding ones instead. Surprisingly, the vector penalty $u_{\mathcal{U}}^{\pi} \in \mathbb{R}^{\mathcal{S}}$ additionally illustrates that the worst kernel is a rank-one perturbation of the nominal. Indeed, considering the stochastic matrix induced by any policy $\pi \in \Pi$, we have

$$[P_{\mathcal{U}}^{\pi} - P_0^{\pi}](s'|s) = - \left( \sum_{a \in \mathcal{A}} \beta_{s,a} \pi_s(a) \right) u_{\mathcal{U}}^{\pi}(s'), \quad \forall s \in \mathcal{S},$$

so that the perturbation matrix $P_{\mathcal{U}}^{\pi} - P_0^{\pi}$ is of rank one. In the sequel, we will leverage this finding to compute the robust visitation frequency.

**Theorem 4.3** ($s$-rectangular case). *Given uncertainty set $\mathcal{U} = \mathcal{U}_p^s$ and any policy $\pi \in \Pi$, the worst model is related to the nominal one through:*

$$R_{\mathcal{U}}^{\pi}(s, a) = R_0(s, a) - \alpha_s \left( \frac{\pi_s(a)}{\|\pi_s\|_q} \right)^{q-1} \qquad and \quad P_{\mathcal{U}}^{\pi}(\cdot|s, a) = P_0(\cdot|s, a) - \beta_s u_{\mathcal{U}}^{\pi} \left( \frac{\pi_s(a)}{\|\pi_s\|_q} \right)^{q-1}.$$

Similarly to the $(s, a)$-case, the adversarial kernel is a rank-one perturbation of the nominal. Yet, an extra dependence on the policy through the coefficient $\left( \frac{\pi_s(a)}{\|\pi_s\|_q} \right)^{q-1}$ appears in the $s$-case, affecting both the worst reward and the worst kernel. Intuitively, it means that the worst model cannot be chosen independently for each action, but must instead depend on the agent's policy. This further explains why optimal policies can all be stochastic in $s$-rectangular robust MDPs [33].

Thms. 4.2 and 4.3 enable us to derive the worst MDP model in closed form with time complexity $O(S^2 A \log \epsilon^{-1})$, up to logarithmic factors (please see the appendix for a detailed discussion). It thus holds the same complexity as non-robust value iteration, since we additionally need to compute the value function to derive its corresponding regularizer [3, 14]. On the other hand, if we employ convex optimization using value methods instead, obtaining the worst model requires a time complexity of $O(S^4 A \log \epsilon^{-1})$ in the $(s, a)$-rectangular case, and $O(S^4 A^3 \log \epsilon^{-1})$ in the $s$-rectangular case [30][Sec. 4.1].

## 4.2 Robust Occupation Measure

We finally derive the robust occupation measure using nominal values, which will lead to an explicit RPG. Although intractable in general, we show that focusing on ball-constrained uncertainty enables deriving the robust occupation matrix efficiently from the (nominal) occupation measure. We first establish the lemma below, which leverages the fact that the worst transition function is a rank-one perturbation of the nominal and represents our core contribution.

**Lemma 4.4.** *Let $b, k \in \mathbb{R}^{\mathcal{S}}$ and $P_0, P_1 \in (\Delta_{\mathcal{S}})^{\mathcal{S}}$ two transition matrices. If $P_1 = P_0 - bk^{\top}$, i.e., $P_1$ is a rank-one perturbation of $P_0$, then their occupation matrices $D_i := (I - \gamma P_i)^{-1}, i = 0, 1$ are related through:*

$$D_1 = D_0 - \gamma \frac{D_0 b k^{\top} D_0}{(1 + \gamma k^{\top} D_0 b)}.$$

Combining Thms. 4.2 and 4.3 with the above lemma, we obtain the robust occupation in terms of the nominal, as stated in Thm. 4.6 below. Prior to this, we introduce the notion of *expected transition uncertainty* below.

**Definition 4.5.** *Let $\mathcal{U}$ a rectangular $\ell_p$-ball-constrained uncertainty set of transition radius $\beta$. For any policy $\pi \in \Pi$, the expected transition uncertainty at any state $s \in \mathcal{S}$ is given by $\beta_s^{\pi} := \sum_{a \in \mathcal{A}} \pi_s(a) \beta_{s,a}$ if $\mathcal{U} = \mathcal{U}_p^{sa}$, and $\beta_s^{\pi} := \beta_s \|\pi_s\|_q$ if $\mathcal{U} = \mathcal{U}_p^s$.*

**Theorem 4.6.** *For any rectangular $\ell_p$-ball-constrained uncertainty and $\pi \in \Pi$, it holds that:*

$$d_{\mathcal{U},\mu}^{\pi} = d_{P_0,\mu}^{\pi} - \gamma \frac{\langle d_{P_0,\mu}^{\pi}, \beta^{\pi} \rangle_{\mathcal{S}}}{1 + \gamma \langle d_{P_0,u_{\mathcal{U}}^{\pi}}^{\pi}, \beta^{\pi} \rangle_{\mathcal{S}}} d_{P_0,u_{\mathcal{U}}^{\pi}}^{\pi}. \tag{3}$$

Thm. 4.6 explicitly highlights the relationship between the robust visitation frequency and the nominal one. Thus, according to Eq. (3), the standard non-robust occupation measure in the first term needs to be penalized by another one, $d_{P_0,u_{\mathcal{U}}^{\pi}}^{\pi} = (u_{\mathcal{U}}^{\pi})^{\top}(I_{\mathcal{S}} - \gamma P_0^{\pi})^{-1}$, to obtain the robust occupation measure. Recall that $u_{\mathcal{U}}^{\pi}$ is the balanced-scaled value function determined by $\pi \in \Pi$ and uncertainty set $\mathcal{U}$. Thus, the penalty term $d_{P_0,u_{\mathcal{U}}^{\pi}}^{\pi}$ tends to zero if all coordinates of the robust value function vector converge to the same value.

Nonetheless, our expression (3) does present some challenges. First, the visitation frequency appearing in the correction term indicates that instead of taking a fixed initial state distribution, we should start from a *varying* and *signed* measure represented by the balanced value function. Although it suggests putting more weight on worst-performing states, obtaining a non-biased estimator for this occupancy measure remains unclear in model-free learning. One may use importance sampling, but as any off-policy approach, both variance and bias would need to be controlled then. Such statistical analysis goes beyond the scope of this work.

## 4.3 Robust Q-values

In this section, we focus on the last element needed for RPG and aim to estimate the robust Q-value denoted previously by $Q_{\mathcal{U}}^{\pi} := Q_{(P_{\mathcal{U}}^{\pi}, R_{\mathcal{U}}^{\pi})}^{\pi}$. Define its associated value function as $v_{\mathcal{U}}^{\pi}(s) = \langle \pi_s, Q_{\mathcal{U}}^{\pi}(s, \cdot) \rangle, \forall s \in \mathcal{S}, \pi \in \Pi$. Based on standard Bellman recursion, it thus holds that:

$$Q_{\mathcal{U}}^{\pi}(s, a) = R_{\mathcal{U}}^{\pi}(s, a) + \gamma \langle P_{\mathcal{U}}^{\pi}(\cdot | s, a), v_{\mathcal{U}}^{\pi} \rangle_{\mathcal{S}}, \quad \forall (s, a) \in \mathcal{X}, \pi \in \Pi,$$

while $Q_{\mathcal{U}}^{\pi}$ is the unique fixed point of the $\gamma$-contracting operator

$$(\mathcal{L}_{\mathcal{U}}^{\pi} Q)(s, a) := T_{(P_{\mathcal{U}}^{\pi}, R_{\mathcal{U}}^{\pi})}^{\pi} Q(s, a), \quad \forall Q \in \mathbb{R}^{\mathcal{X}}. \tag{4}$$

The relations above hold for general uncertainty sets, provided that we have access to the worst model. The $s$-rectangularity assumption additionally enables us to retrieve the robust value function using the Bellman operator above [33]. Concretely, we have: $v_{\mathcal{U}}^{\pi} = \min_{(P,R) \in \mathcal{U}} v_{(P,R)}^{\pi} = v_{(P_{\mathcal{U}}^{\pi}, R_{\mathcal{U}}^{\pi})}^{\pi}$.

The following result derives a regularized operator equivalent to $\mathcal{L}_{\mathcal{U}}^{\pi}$, which results in an efficient iteration method to compute the robust Q-value.

**Proposition 4.7.** *The Bellman operator $\mathcal{L}_{\mathcal{U}}^{\pi}$ defined in Eq. (4) is equivalent to:*

$$(\mathcal{L}_{\mathcal{U}}^{\pi} Q)(s, a) = T_{(P_0, R_0)}^{\pi} Q(s, a) + \Omega_q'(\alpha_{s,a}, \beta_{s,a}, v),$$

*where $v(s) := \langle \pi_s, Q(s, \cdot) \rangle_{\mathcal{A}}$, $\Omega_q'(\alpha, \beta, v) := -(\alpha_{s,a} + \gamma \beta_{s,a} \kappa_q(v))$ for $(s, a)$-rectangular uncertainty $\mathcal{U}_p^{sa}$, and $\Omega_q'(\alpha, \beta, v) := -\left(\frac{\pi_s(a)}{\|\pi_s\|_q}\right)^{q-1}(\alpha_s + \gamma \beta_s \kappa_q(v))$ for s-rectangular $\mathcal{U}_p^s$.*

# 5   Robust Policy Gradient

We are now able to derive an RPG by combining our previous results. Notably, unlike previous works that need to sample next-state transitions based on all models from the uncertainty set [21, 17, 4], here, we only need the nominal kernel to get the occupation measures.

**Theorem 5.1** (RPG). *For any rectangular $\ell_p$-ball-constrained uncertainty, the robust policy gradient is given by:*

$$\partial_\pi \rho_{\mathcal{U}}^\pi = \sum_{(s,a)\in\mathcal{X}} \left(d_{P_0,\mu}^\pi(s) - c^\pi(s)\right) Q_{\mathcal{U}}^\pi(s,a)\nabla\pi_s(a), \tag{5}$$

*where*

$$c^\pi(s) := \frac{\gamma\langle d_{P_0,\mu}^\pi, \beta^\pi\rangle_{\mathcal{S}}}{1 + \gamma\langle d_{P_0,u_{\mathcal{U}}^\pi}^\pi, \beta^\pi\rangle_{\mathcal{S}}} d_{P_0,u_{\mathcal{U}}^\pi}^\pi(s), \quad \forall s \in \mathcal{S}\,.$$

The implementation of RPG directly follows and can be found in Alg. 1. Thm. 5.1 is a straightforward application of non-robust PG, as its proof simply consists in plugging Eq. (3) into the standard PG expression $\partial_\pi \rho_{\mathcal{U}}^\pi = \sum_{(s,a)\in\mathcal{X}} d_{\mathcal{U},\mu}^\pi(s)Q_{\mathcal{U}}^\pi(s,a)\nabla\pi_s(a)$. We obtain a regular PG in the first term, with the robust Q-value instead of the non-robust one, plus a correction term $c^\pi$ resulting from taking the visitation frequency of the worst kernel instead of the nominal. Unlike previous work that uses policy regularization to achieve empirical robustness in PG methods [2, 11], Thm. 5.1 establishes an RPG that accounts for transition uncertainty and targets a robust optimal policy. It crucially relies on the rank-one-perturbation structure of the worst transition kernel (see Lem. 4.4). As established in Thm. 4.6, $\ell_p$-ball uncertainty implies such property, but the converse as to whether any convex set leads to the worst transition kernel being a rank-one perturbation of the nominal remains an open question. For example, it would be interesting to investigate the structural properties needed on the uncertainty set for the rank-one perturbation to hold.

---

**Algorithm 1** RPG

> **Input:** $\mu, \eta$
> **Initialize:** $v_k, \pi_k$
> 1: **for** $k = 1, 2, \ldots$ **do**
> 2: $\quad \partial_\pi \rho_k = \sum_{(s,a)\in\mathcal{X}} \left(d_{P_0,\mu}^\pi(s) - c^\pi(s)\right) Q_{\mathcal{U}}^\pi(s,a)\nabla\pi_s(a)$ $\qquad\qquad$ ▷ Compute policy gradient
> 3: $\quad \pi_k \leftarrow \mathbf{proj}_\Pi(\pi_k + \eta\partial_\pi\rho_k)$ $\qquad\qquad\qquad\qquad\qquad\qquad\qquad$ ▷ Update policy
> 4: **end for**

---

## 5.1   Complexity Analysis

A major concern in solving robust MDPs is time complexity [33]. Similarly, it is of major importance to assess the additional time required for computing an RPG update, compared to its non-robust variant. Although previous work has analyzed the convergence rate of RPG to a global optimum [30], it assumes access to an oracle gradient, thus occulting the computational concerns raised from gradient estimation. In fact, the NP-hardness of non-rectangular and/or non-convex robust MDPs [33] already indicates that their resulting RPG can be intractable.

To compute RPG in Thm. 5.1, we first need to evaluate the robust Q-value. Based on Lemma 4.7 and the Bellman operators introduced there, our evaluation method involves an additional estimation of the variance function $\kappa_p$. According to [14], this takes logarithmic time at most, using binary search. As to the compensation term $c^\pi$ in Eq. (11), it requires computing occupancy measures with respect to two different initial vectors, namely the balanced value function and the initial distribution. Thus, the computational cost for estimating $c^\pi$ is the same as estimating a non-robust occupancy measure. Tab. 1 summarizes the complexity of different approaches while a detailed discussion can be found in the appendix. We refer to [30][Sec. 4.1] for the complexity of RPG based on convex optimization.

**Generalization to arbitrary norms.**   Until now, we have focused on $\ell_p$-norm for concreteness. However, the above results apply to any norm $\|\cdot\|$, at least if the uncertainty set is $(s,a)$-rectangular, in which case the variance function changes to $\kappa(v) := \min_{\|c\|\le 1,\mathbf{1}^\top c=0}\langle c, v\rangle$ and the balanced value to $\arg\min_{\|c\|\le 1,\mathbf{1}^\top c=0}\langle c, v\rangle$. The rank-one perturbation structure of the worst kernel is preserved, so the robust occupation measure can be obtained similarly using Lemma 4.4. The $s$-rectangular is more involved. We defer its discussion to the appendix and leave its complete derivation for future work.

## 6 Experiments

In order to test the effectiveness of our RPG update, we evaluate its increased time complexity relative to non-robust PG. In the following experiments, we randomly generate nominal models for arbitrary state-action space sizes. Each experiment was averaged over 100 runs. We refer the reader to the appendix for more details on the radius levels and other implementation choices.

We first focus on $\ell_1$-robust MDPs to compare our RPG with a convex optimization approach. Specifically, we consider a robust PG with an optimization solver, which we designate by LP-RPG. Indeed, recall that $\ell_1$-ball-constraints induce a linear program (LP) rather than a more general convex optimization problem. Therefore, to compute the robust value function for a given policy, we iteratively evaluate the robust Bellman operator using LP [30, Section 4.1]. Using this approximated value function, we can compute the worst value parameters to apply PG theorem by [26] and deduce an LP-based robust PG update. Differently, our RPG method relies on the regularized formulation of robust value iteration proposed in [3, 14], from which we deduce the normalized-balanced value function as in Eq. (2). We finally apply Thm. 4.6 to compute the robust occupation measure, and Prop. 4.7 to obtain the robust Q-value.

Tab. 3 displays the results obtained for the two alternative methods described above. In all experiments, the standard deviation was typically 2-10% so we omitted it for brevity. As can be seen in Tab. 3, LP-RPG does not scale well compared to RPG, whereas RPG has similar time complexity as PG. Notably, the running time of $s$-rectangular LP-RPG scales much better with the space size than its $(s, a)$-rectangular equivalent, which confirms the theoretical complexities from Tab. 1. Yet, since these methods were time-consuming, we repeated these for a few runs only. In fact, LP-RPG is more expensive than RPG by 1-3 orders of magnitude, which illustrates its inefficiency. We emphasize that here, we only focused on $\ell_1$-robust MDPs to leverage LP solvers in robust policy evaluation. We expect the computational cost of LP-RPG to scale even more poorly for other $\ell_p$-robust MDPs that involve polynomial time-consuming convex programs.

Table 3: Comparison of the relative running time between RPG and the convex optimization approach (here, LP). Our method is faster than LP-based updates by 1 to 3 orders of magnitude.

| | | $\|\ \{(P_0, R_0)\}\ \|$ | $\mathcal{U}_1^{\mathsf{sa}}$ | | $\mathcal{U}_1^{\mathsf{s}}$ | |
|---|---|---|---|---|---|---|
| S | A | PG | RPG | LP-RPG | RPG | LP-RPG |
| 10 | 10 | 1 | 1.4 | 326 | 1.4 | 77 |
| 30 | 10 | 1 | 1.4 | 351 | 1.4 | 109 |
| 50 | 10 | 1 | 1.4 | 408 | 1.4 | 159 |
| 100 | 20 | 1 | 1.5 | 469 | 1.3 | 268 |
| 500 | 50 | 1 | 1.3 | 925 | 1.3 | 5343 |

We further compare our RPG to non-robust PG on different $\ell_p$-balls. Tab. 4 confirms the comparable time complexity of RPG to non-robust PG, thus demonstrating the effectiveness of our method. We note that for $p \in \{1, 2, \infty\}$, the corresponding regularization quantities can be computed in closed form, whereas they involve a binary search for other values [14]. We thus get a slight running-time increase for $p \in \{5, 10\}$.

Table 4: Relative running time for computing RPG under different types of uncertainty sets.

| S | A | $\{(P_0, R_0)\}$ | $\mathcal{U}_2^{\mathsf{sa}}$ | $\mathcal{U}_2^{\mathsf{s}}$ | $\mathcal{U}_5^{\mathsf{sa}}$ | $\mathcal{U}_5^{\mathsf{s}}$ | $\mathcal{U}_{10}^{\mathsf{sa}}$ | $\mathcal{U}_{10}^{\mathsf{s}}$ | $\mathcal{U}_{\infty}^{\mathsf{sa}}$ | $\mathcal{U}_{\infty}^{\mathsf{s}}$ |
|---|---|---|---|---|---|---|---|---|---|---|
| 10 | 10 | 1 | 1.5 | 1.5 | 4.9 | 4.7 | 4.7 | 4.9 | 1.5 | 1.6 |
| 30 | 10 | 1 | 1.4 | 1.5 | 4.2 | 4.3 | 4.2 | 4.0 | 1.4 | 1.4 |
| 50 | 10 | 1 | 1.5 | 1.4 | 4.5 | 4.1 | 4.0 | 4.0 | 1.4 | 1.4 |
| 100 | 20 | 1 | 1.4 | 1.3 | 2.6 | 2.5 | 2.5 | 2.4 | 1.3 | 1.2 |
| 500 | 50 | 1 | 1.2 | 1.2 | 1.7 | 1.7 | 1.7 | 1.7 | 1.2 | 1.3 |

## 7 Discussion

This paper introduced an explicit expression of RPG for rectangular robust MDPs. Our approach involved auxiliary results such as deriving the worst model in closed form and showing that it is a rank-one perturbation of the nominal kernel. The resulting RPG extends vanilla PG with additional correction terms that can be derived in closed form as well. Thus, the computational time of RPG is similar to its non-robust variant.

A key assumption that would be interesting to relax is the normed-ball structure of the uncertainty sets considered in this study. Indeed, since the proofs of our technical results rely on norm properties, it is still unclear if and how RPG can generalize to metric-based or $f$-divergence uncertainty sets. The latter type of uncertainty can be particularly useful for data-driven settings, as the radius can be chosen according to cross-validation or statistical bounds [10]. Another compelling direction would be to explore other variants of RPG using mirror descent or natural policy gradient and examine their compatibility with deep architectures, which would further demonstrate the practical efficiency of our RPG method.

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

## Contents

# A  Balanced and Normed Vectors

In this section, we lay down some basic properties of $p$-normalized-balanced vectors.

First recall the $p$-variance and the $p$-mean defined as:

$$\kappa_p(v) = \min_{\omega \in \mathbb{R}} \|v - \omega \mathbf{1}\|_p, \qquad \omega_p(v) = \arg \min_{\omega \in \mathbb{R}} \|v - \omega \mathbf{1}\|_p.$$

Given any $v \in \mathbb{R}^{\mathcal{S}}$, let also the $p$-balanced-normalized function:

$$u_p(v)(s) := \text{SIGN}(v(s) - \omega_p(v)) \left( \frac{|v(s) - \omega_p(v)|}{\kappa_p(v)} \right)^{p-1}, \quad \forall v \in \mathbb{R}^{\mathcal{S}}, s \in \mathcal{S}.$$

According to [14][Sec. 16.1, Lemma 1], the following holds:

$$\kappa_q(v) = -\frac{1}{\epsilon} \left[ \min_{\|c\|_p \leq \epsilon, \langle c, \mathbf{1} \rangle = 0} \langle c, v \rangle \right], \tag{6}$$

namely, the $p$-variance function is the optimal value of a linear optimization under kernel noise constraint. The result below further characterizes the solution to the above problem.

**Lemma A.1.** *The vector defined as $c^* := -\epsilon u_q(v)$ is an optimal solution to the optimization problem*

$$\min_{\|c\|_p \leq \epsilon, \langle c, \mathbf{1} \rangle_{\mathcal{S}} = 0} \langle c, v \rangle.$$

*Proof.* It suffices to show that $c^*$ satisfies both constraints $\|c^*\|_p \leq \epsilon$ and $\langle c^*, \mathbf{1} \rangle_{\mathcal{S}} = 0$, and that it reaches optimal value, i.e., $-\frac{1}{\epsilon}\langle c^*, v \rangle = \kappa_q(v)$. We thus compute:

$$\|c^*\|_p = \left( \sum_{s \in \mathcal{S}} |c^*(s)|^p \right)^{\frac{1}{p}}$$

$$= \left( \sum_{s \in \mathcal{S}} \left| -\epsilon \text{SIGN}(v(s) - \omega_q(v)) \left( \frac{|v(s) - \omega_q(v)|}{\kappa_q(v)} \right)^{q-1} \right|^p \right)^{\frac{1}{p}}$$

$$= \left( \left( \frac{\epsilon}{\kappa_q(v)^{q-1}} \right)^p \sum_{s \in \mathcal{S}} \left| \left( \frac{|v(s) - \omega_q(v)|}{\kappa_q(v)} \right)^{q-1} \right|^p \right)^{\frac{1}{p}}$$

$$= \frac{\epsilon}{\kappa_q(v)^{q-1}} \left( \sum_{s \in \mathcal{S}} |v(s) - \omega_q(v)|^{(q-1)p} \right)^{\frac{1}{p}}$$

$$= \frac{\epsilon}{\kappa_q(v)^{q-1}} \left( \sum_{s \in \mathcal{S}} |v(s) - \omega_q(v)|^q \right)^{\frac{1}{p}} \qquad \text{(By assumption, } \frac{p+q}{pq} = 1\text{)}$$

$$= \frac{\epsilon}{\kappa_q(v)^{q-1}} \kappa_q(v)^{\frac{q}{p}} \qquad \text{(By definition, } \kappa_q(v) = \|v - \omega_q \mathbf{1}\|_q\text{)}$$

$$= \epsilon, \qquad (\frac{q}{p} - (q-1) = \frac{q - pq + p}{p} = 0)$$

so the norm constraint is satisfied. We check the noise constraint by computing:

$$\sum_{s \in \mathcal{S}} c^*(s) = \sum_{s \in \mathcal{S}} -\epsilon \text{SIGN}(v(s) - \omega_q(v)) \left( \frac{|v(s) - \omega_q(v)|}{\kappa_q(v)} \right)^{q-1}$$

$$= \frac{-\epsilon}{\kappa_q(v)^{q-1}} \sum_{s \in \mathcal{S}} \text{SIGN}(v(s) - \omega_q(v)) |v(s) - \omega_q(v)|^{q-1}.$$

Now, considering the real function $\varphi : w \to \|v - w\mathbf{1}\|_q$ and taking its derivative, we remark the proportional relation:

$$\sum_{s \in \mathcal{S}} c^*(s) = C \cdot \varphi'(\omega_q(v)),$$

where $C \in \mathbb{R}$ is the proportionality coefficient. By construction, $\omega_q(v)$ is a minimizer of $\varphi$, so we must have $\varphi'(\omega_q(v)) = 0$ and $c^*$ satisfies the noise constraint.

We finally show that $c^*$ reaches the optimal value:

$$
\begin{aligned}
-\frac{1}{\epsilon}\langle c^*, v\rangle &= -\frac{1}{\epsilon}\langle c^*, v - \omega_q(v)\mathbf{1}\rangle & (\langle c^*, \mathbf{1}\rangle_{\mathcal{S}} = 0) \\
&= \sum_{s\in\mathcal{S}} \frac{|v(s) - \omega_q(v)|^q}{\kappa_q(v)^{q-1}} & \text{(Putting the value of } c^*) \\
&= \frac{\kappa_q(v)^q}{\kappa_q(v)^{q-1}} & (\kappa_q(v) = \|v - \omega_q\mathbf{1}\|_q) \\
&= \kappa_q(v).
\end{aligned}
$$

$\square$

### A.1 Proof of Proposition 4.1

**Proposition.** *For any policy $\pi \in \Pi$ and $\ell_p$-ball rectangular uncertainty set, the following holds:*

$$
u_{\mathcal{U}}^\pi = \nabla_v \kappa_q(v)\Big|_{v=v_{\mathcal{U}}^\pi},
$$
$$
\langle u_{\mathcal{U}}^\pi, v_{\mathcal{U}}^\pi\rangle = \kappa_q(v_{\mathcal{U}}^\pi).
$$

*Proof.* The second claim directly follows from Lemma A.1 applied to $v := v_{\mathcal{U}}^\pi$, so that by optimality, $\kappa_q(v_{\mathcal{U}}^\pi) = \langle u_{\mathcal{U}}^\pi, v_{\mathcal{U}}^\pi\rangle$. For the first claim, we take the gradient of $\kappa_p(v) := \min_{w\in\mathbb{R}}\|v - w\mathbf{1}\|_p$ w.r.t. $v$ using the envelope theorem [19]. Then, the $p$-balanced-normalized vector $u_p(v)$ is a sub-gradient of $\kappa_p(v)$, that is,

$$
u_p(v) = \nabla\kappa_q(v),
$$

which we apply to $v := v_{\mathcal{U}}^\pi$. $\square$

We have the additional properties below:

- The variance function $\kappa_q$ is translation-invariant in all-ones directions, i.e., for all $\omega \in \mathbb{R}, \kappa_q(v) = \kappa_q(v + \omega\mathbf{1})$. As a result, $\langle\nabla\kappa_q(v), \mathbf{1}\rangle_{\mathcal{S}} = 0$.
- The balanced-normalized vector $u_p(v)$ has unit norm, i.e., $\|u_p(v)\|_p = 1$ by Lemma A.1.

## B   Worst Kernel and Reward

Here we present the proofs for the worst/adversarial kernel and reward function characterization.

### B.1 Proof of Theorem 4.2

**Theorem** (($s, a$)-rectangular case). *Given uncertainty set $\mathcal{U} = \mathcal{U}_p^{sa}$ and any policy $\pi \in \Pi$, the worst model is related to the nominal one through:*

$$
R_{\mathcal{U}}^\pi(s, a) = R_0(s, a) - \alpha_{s,a} \qquad \text{and} \qquad P_{\mathcal{U}}^\pi(\cdot|s, a) = P_0(\cdot|s, a) - \beta_{s,a}u_{\mathcal{U}}^\pi.
$$

*Proof.* By definition,

$$
(P_{\mathcal{U}_p^{sa}}^\pi, R_{\mathcal{U}_p^{sa}}^\pi) \in \underset{(P,R)\in\mathcal{U}_p^{sa}}{\arg\min}\, T_{(P,R)}^\pi v_{\mathcal{U}_p^{sa}}^\pi.
$$

Additionally, since $\mathcal{U}_p^{sa} = (R_0 + \mathcal{R}) \times (P_0 + \mathcal{P})$, it results that:

$$
(R_{\mathcal{U}_p^{sa}}^\pi, P_{\mathcal{U}_p^{sa}}^\pi) = (P_0 + P^*, R_0 + R^*)
$$

where

$$
(P^*, R^*) \in \underset{(P,R)\in\mathcal{P}\times\mathcal{R}}{\arg\min}\, T_{(P,R)}^\pi v_{\mathcal{U}_p^{sa}}^\pi.
$$

By the $(s, a)$-rectangularity assumption, we get that for all $(s, a) \in \mathcal{X}$,

$$(P^*(\cdot|s,a), R^*(s,a)) \in \underset{(p_{s,a}, r_{s,a}) \in \mathcal{P}_{s,a} \times \mathcal{R}_{s,a}}{\arg \min} \left\{ r_{s,a} + \gamma \sum_{s' \in \mathcal{S}} p_{s,a}(s') v_{\mathcal{U}_p^\pi}^\pi(s') \right\}$$

It is clear from the above that the worst reward is independent of policy $\pi$. Thus, by the ball constraint, it is given by

$$R^*(s,a) = -\alpha_{s,a}, \quad \forall(s,a) \in \mathcal{X}.$$

Differently, the worst kernel depends on the value function which itself depends on the policy. It is given by

$$P^*(\cdot|s,a) = \underset{p_{s,a} \in \mathcal{P}_{sa}}{\arg \min} \left\{ \sum_{s' \in \mathcal{S}} p_{s,a}(s') v_{\mathcal{U}_p^\pi}^\pi(s') \right\}, \quad \forall(s,a) \in \mathcal{X}.$$

The optimization is of the form

$$\underset{\|c\|_p \leq \beta, \langle c, \mathbf{1} \rangle = 0}{\arg \min} \langle c, v \rangle,$$

so by Lemma A.1,

$$P^*(s'|s,a) = -\beta_{s,a} \text{SIGN}\left( v_{\mathcal{U}_p^\pi}^\pi(s') - \omega_q(v_{\mathcal{U}_p^\pi}^\pi) \right) \frac{\left| v_{\mathcal{U}_p^\pi}^\pi(s') - \omega_q(v_{\mathcal{U}_p^\pi}^\pi) \right|^{q-1}}{\kappa_q(v)^{q-1}}.$$

As a result, we proved that for all $(s, a) \in \mathcal{X}$, $R_{\mathcal{U}_p^\pi}^\pi(s,a) = R_0(s,a) - \alpha_{s,a}$ and

$$P_{\mathcal{U}_p^\pi}^\pi(s'|s,a) = P_0(s'|s,a) - \beta_{s,a} \text{SIGN}\left( v_{\mathcal{U}_p^\pi}^\pi(s') - \omega_q(v_{\mathcal{U}_p^\pi}^\pi) \right) \frac{\left| v_{\mathcal{U}_p^\pi}^\pi(s') - \omega_q(v_{\mathcal{U}_p^\pi}^\pi) \right|^{q-1}}{\kappa_q(v)^{q-1}}.$$

$\square$

## B.2 Proof of Theorem 4.3

**Theorem** (*s-rectangular case*). *Given uncertainty set $\mathcal{U} = \mathcal{U}_p^s$ and any policy $\pi \in \Pi$, the worst model is related to the nominal one through:*

$$R_{\mathcal{U}}^\pi(s,a) = R_0(s,a) - \alpha_s \left( \frac{\pi_s(a)}{\|\pi_s\|_q} \right)^{q-1} \quad and \quad P_{\mathcal{U}}^\pi(\cdot|s,a) = P_0(\cdot|s,a) - \beta_s u_{\mathcal{U}}^\pi \left( \frac{\pi_s(a)}{\|\pi_s\|_q} \right)^{q-1}.$$

*Proof.* By definition,

$$(P_{\mathcal{U}_p^s}^\pi, R_{\mathcal{U}_p^s}^\pi) \in \underset{(P,R) \in \mathcal{U}_p^s}{\arg \min} T_{(P,R)}^\pi v_{\mathcal{U}_p^s}^\pi,$$

and since $\mathcal{U}_p^s = (R_0 + \mathcal{R}) \times (P_0 + \mathcal{P})$, we have

$$(P_{\mathcal{U}_p^s}^\pi, R_{\mathcal{U}_p^s}^\pi) = (P_0 + P^*, R_0 + R^*)$$

where

$$(P^*, R^*) \in \underset{(P,R) \in \mathcal{P} \times \mathcal{R}}{\arg \min} T_{(P,R)}^\pi v_{\mathcal{U}_p^{sa}}^\pi.$$

By the $s$-rectangularity assumption, we get that for all $s \in \mathcal{S}$

$$(P^*(\cdot|s,\cdot), R^*(s,\cdot)) = \underset{(p_s, r_s) \in \mathcal{P}_s \times \mathcal{R}_s}{\arg \min} \sum_{a \in \mathcal{A}} \pi_s(a) \left\{ r_{s,a} + \gamma \sum_{s' \in \mathcal{S}} p_{s,a}(s') v_{\mathcal{U}_p^{sa}}^\pi(s') \right\}.$$

Here, the worst reward does depend on policy $\pi$ and is given by

$$R^*(s,a) = -\alpha_s \frac{\pi_s(a)^{q-1}}{\sum_a \pi_s(a)^{q-1}}, \quad \forall(s,a) \in \mathcal{X}.$$

As for the worst kernel, it depends both on the value function and the policy. It is given by

$$P^*(\cdot|s,\cdot) = \underset{p_s \in \mathcal{P}_s}{\arg\min} \left\{ \sum_{a \in \mathcal{A}} \pi_s(a) \sum_{s' \in \mathcal{S}} p_{s,a}(s') v_{\mathcal{U}_p^\pi}^\pi(s') \right\}.$$

The optimization of interest is of the form

$$\underset{\|c_a\|_p \leq \beta_s, \langle c_a, \mathbf{1} \rangle = 0, a \in \mathcal{A}}{\min} \left\{ \sum_{a' \in \mathcal{A}} \pi_s(a') \langle c_{a'}, v \rangle \right\},$$

which is equivalent to the following two-fold minimization:

$$\underset{\sum_{a \in \mathcal{A}} (\beta_{s,a})^p \leq (\beta_s)^p}{\min} \quad \underset{\|c_a\|_p \leq \beta_s, \langle c_a, \mathbf{1} \rangle = 0, a \in \mathcal{A}}{\min} \left\{ \sum_{a' \in \mathcal{A}} \pi_s(a') \langle c_{a'}, v \rangle \right\}.$$

Thus, rewriting the problem in our context,

$$\underset{\sum_a (\beta_{s,a})^p \leq (\beta_s)^p}{\min} \quad \underset{\|p_{sa}\|_p \leq \beta_{s,a}, \sum_{s'} p_{sa}(s') = 0}{\min} \quad \sum_a \pi_s(a) \langle p_{s,a}, v \rangle$$

$$= \underset{\sum_a (\beta_{s,a})^p \leq (\beta_s)^p}{\min} \sum_a \pi_s(a) \underset{\|p_{sa}\|_p \leq \beta_{s,a}, \sum_{s'} p_{sa}(s') = 0}{\min} \quad \langle p_{s,a}, v \rangle$$

$$= \underset{\sum_a (\beta_{sa})^p \leq (\beta_s)^p}{\min} \sum_a \pi_s(a)(-\beta_{sa} \kappa_q(v)) \qquad \text{(By Lemma A.1)}$$

$$= -\kappa_q(v) \underset{\sum_a (\beta_{sa})^p \leq (\beta_s)^p}{\max} \sum_a \pi_s(a) \beta_{sa}.$$

Computing the optimal $\beta$ above, the optimization is now the same as in the $(s,a)$-rectangular case. Hence, we have

$$P^*(s'|s,a) = -\beta_s \frac{\pi_s(a)^{q-1}}{\|\pi_s\|_q^{q-1}} \text{SIGN}(v_{\mathcal{U}_p^\pi}^\pi(s') - \omega_q(v_{\mathcal{U}_p^\pi}^\pi)) \frac{\left| v_{\mathcal{U}_p^\pi}^\pi(s') - \omega_q(v_{\mathcal{U}_p^\pi}^\pi) \right|^{q-1}}{\kappa_q(v)^{q-1}},$$

which ends the proof by definition of the balanced value function $u_{\mathcal{U}}^\pi$. □

## C Occupation Matrix

### C.1 Proof of Lemma 4.4

**Lemma.** *Let $b, k \in \mathbb{R}^{\mathcal{S}}$ and $P_0, P_1 \in (\Delta_{\mathcal{S}})^{\mathcal{S}}$ two transition matrices. If $P_1 = P_0 - bk^\top$, i.e., $P_1$ is a rank-one perturbation of $P_0$, then their occupation matrices $D_i := (I - \gamma P_i)^{-1}, i = 0, 1$ are related through:*

$$D_1 = D_0 - \gamma \frac{D_0 b k^\top D_0}{(1 + \gamma k^\top D_0 b)}.$$

*Proof.* By definition, $D_1 = (I_{\mathcal{S}} - \gamma P_1)^{-1}$ so it follows that:

$$(I_{\mathcal{S}} - \gamma P_1) D_1 = I_{\mathcal{S}}$$
$$\iff I_{\mathcal{S}} + \gamma P_1 D_1 = D_1$$
$$\iff I_{\mathcal{S}} + \gamma(P_0 - bk^\top) D_1 = D_1 \qquad \text{(By assumption, } P_1 = P_0 - bk^\top\text{)}$$
$$\iff I_{\mathcal{S}} - \gamma bk^\top D_1 = (I_{\mathcal{S}} - \gamma P_0) D_1$$
$$\iff (I_{\mathcal{S}} - \gamma P_0)^{-1}(I_{\mathcal{S}} - \gamma bk^\top D_1) = D_1 \qquad \text{(Multiplying both sides by } (I_{\mathcal{S}} - \gamma P_0)^{-1}\text{)}$$
$$\iff D_0(I_{\mathcal{S}} - \gamma bk^\top D_1) = D_1 \qquad \text{(By definition, } D_0 = (I_{\mathcal{S}} - \gamma P_0)^{-1}\text{)}$$
$$\iff D_0 - \gamma D_0 bk^\top D_1 = D_1. \qquad (7)$$

Now, multiplying both sides by $k$ and noticing that $k^\top D_0 b$ is a scalar we get

$$k^\top D_0 - \gamma k^\top D_0 b k^\top D_1 = k^\top D_1$$
$$\iff k^\top D_0 = (1 + \gamma k^\top D_0 b) k^\top D_1$$
$$\iff k^\top D_1 = \frac{k^\top D_0}{(1 + \gamma k^\top D_0 b)}. \tag{8}$$

Combining Eqs. (7) and (8) thus yields:

$$D_1 = D_0 - \gamma \frac{D_0 b k^\top D_0}{(1 + \gamma k^\top D_0 b)},$$

which concludes the proof. $\qquad\square$

### C.2 Proof of Theorem 4.6

**Theorem.** *For any rectangular $\ell_p$-ball-constrained uncertainty and $\pi \in \Pi$, it holds that:*

$$d^\pi_{\mathcal{U},\mu} = d^\pi_{P_0,\mu} - \gamma \frac{\langle d^\pi_{P_0,\mu}, \beta^\pi \rangle_\mathcal{S}}{1 + \gamma \langle d^\pi_{P_0, u^\pi_\mathcal{U}}, \beta^\pi \rangle_\mathcal{S}} d^\pi_{P_0, u^\pi_\mathcal{U}}.$$

*Proof.* From Thms. 4.2 and 4.3, it holds that:

$$P^\pi_\mathcal{U}(s'|s) = P^\pi_0(s'|s) - \beta^\pi_s u^\pi_\mathcal{U}(s'), \quad \forall s, s' \in \mathcal{S}.$$

Therefore, setting $P_1 := P^\pi_\mathcal{U}$, $P_0 := P^\pi_0$, $b := \beta^\pi$ and $k := u^\pi_\mathcal{U}$, we can apply Lemma 4.4 and relate the corresponding occupation matrices. Additionally multiplying both sides of the relation by $\mu^\top \in \mathbb{R}^{1 \times \mathcal{S}}$ yields the desired result. $\qquad\square$

## D Robust Q-value

### D.1 Basic Properties

In the literature, robust Q-values are defined in various ways that turn out to be conflicting for $s$ but non-$(s, a)$ rectangular uncertainty sets. In this section, we propose to define the robust Q-value solely based on the worst model. Define the robust Q-value, the robust value function, and the robust occupation respectively as:

$$Q^\pi_\mathcal{U} := Q^\pi_{(P^\pi_\mathcal{U}, R^\pi_\mathcal{U})}, \quad d^\pi_\mathcal{U} := d^\pi_{(P^\pi_\mathcal{U}, R^\pi_\mathcal{U})}, \quad v^\pi_\mathcal{U} := v^\pi_{(P^\pi_\mathcal{U}, R^\pi_\mathcal{U})}.$$

For $s$-rectangular uncertainty sets (in particular, for $(s, a)$-rectangular), the above definition of robust value function coincides with the common one, i.e., $v^\pi_{(P^\pi_\mathcal{U}, R^\pi_\mathcal{U})} = \min_{(P,R) \in \mathcal{U}} v^\pi_{(P,R)}$ [33]. If the uncertainty set is additionally $(s, a)$-rectangular (as in [31] or [3, 14]), the above definition of robust Q-value also coincides with the common one because then,

$$Q^\pi_{\mathcal{U}^{\mathrm{sa}}}(s, a) = \min_{(P,R) \in \mathcal{U}^{\mathrm{sa}}} \left( R(s, a) + \gamma \sum_{s' \in \mathcal{S}} P(s'|s, a) v^\pi_{\mathcal{U}^{\mathrm{sa}}}(s') \right), \quad \forall (s, a) \in \mathcal{X}.$$

Getting back to our own definition, robust Q-value and value functions are related through:

$$v^\pi_\mathcal{U}(s) = \langle \pi_s, Q^\pi_\mathcal{U}(s, \cdot) \rangle_\mathcal{A}$$
$$Q^\pi_\mathcal{U}(s, a) = R^\pi_\mathcal{U}(s, a) + \gamma \sum_{s' \in \mathcal{S}} \pi_s(a) P^\pi_\mathcal{U}(s'|s, a) v^\pi_\mathcal{U}(s'),$$

as both quantities are defined based on worst kernel and reward, i.e., $Q^\pi_\mathcal{U} := Q^\pi_{(P^\pi_\mathcal{U}, R^\pi_\mathcal{U})}$ and $v^\pi_\mathcal{U} := v^\pi_{(P^\pi_\mathcal{U}, R^\pi_\mathcal{U})}$.

Given an optimal robust policy $\pi^*_\mathcal{U}$, we further use $P^*_\mathcal{U}, R^*_\mathcal{U}, v^*_\mathcal{U}, Q^*_\mathcal{U}, d^*_\mathcal{U}$ as a shorthand for $P^{\pi^*_\mathcal{U}}_\mathcal{U}, R^{\pi^*_\mathcal{U}}_\mathcal{U}, v^{\pi^*_\mathcal{U}}_\mathcal{U}, Q^{\pi^*_\mathcal{U}}_\mathcal{U}, d^{\pi^*_\mathcal{U}}_\mathcal{U}$ respectively. For $(s, a)$-rectangular uncertainty set $\mathcal{U}^{\mathrm{sa}}$, the optimal value function is the best optimal Q-value, that is

$$v^*_{\mathcal{U}^{\mathrm{sa}}}(s) = \max_{a \in \mathcal{A}} Q^*_{\mathcal{U}^{\mathrm{sa}}}(s, a), \quad \forall s \in \mathcal{S}.$$

because an optimal policy deterministically takes the action with the highest Q-value [20, 12]. This does no longer hold for $s$-rectangular or coupled uncertainty sets, as there, an optimal policy may be stochastic [33]. Still, based on Thms. 4.2 and 4.3, we get the Bellman recursion below.

**Proposition D.1.** *Let an $\ell_p$-ball constrained uncertainty set. Then, for all $(s, a) \in \mathcal{X}$ and $\pi \in \Pi$, the robust Q-value satisfies the following recursion in the $(s, a)$ and $s$-rectangular case respectively:*

$$Q^\pi_{\mathcal{U}^{sa}_p}(s, a) = T^\pi_{(P_0, R_0)} Q^\pi_{\mathcal{U}^{sa}_p}(s, a) - \alpha_{sa} - \gamma \beta_{sa} \kappa_q(v^\pi_{\mathcal{U}^{sa}_p}),$$

$$Q^\pi_{\mathcal{U}^s_p}(s, a) = T^\pi_{(P_0, R_0)} Q^\pi_{\mathcal{U}^s_p}(s, a) - \left( \frac{\pi_s(a)}{\|\pi_s\|_q} \right)^{q-1} \left( \alpha_s + \gamma \beta_s \kappa_q(v^\pi_{\mathcal{U}^s_p}) \right).$$

*Proof.* We give proof for the $(s, a)$-rectangular case only. The $s$-rectangular case follows the exact same lines except that it uses Thm. 4.3 instead of Thm. 4.2. We have:

$$Q^\pi_{\mathcal{U}}(s, a) = Q^\pi_{(P^\pi_{\mathcal{U}}, R^\pi_{\mathcal{U}})}(s, a) \qquad \text{(By definition)}$$

$$= R^\pi_{\mathcal{U}}(s, a) + \sum_{s' \in \mathcal{S}} P^\pi_{\mathcal{U}}(s'|s, a) v^\pi_{\mathcal{U}^{sa}_p}(s')$$

$$= R_0(s, a) - \alpha_{sa} + \gamma \sum_{s' \in \mathcal{S}} \left( P_0(s'|s, a) - \beta_{sa} u^\pi_{\mathcal{U}^{sa}_p}(s') \right) v^\pi_{\mathcal{U}^{sa}_p}(s') \qquad \text{(By Thm. 4.2)}$$

$$= R_0(s, a) - \alpha_{sa} + \gamma \sum_{s' \in \mathcal{S}} P_0(s'|s, a) v^\pi_{\mathcal{U}^{sa}_p}(s') - \gamma \beta_{sa} \kappa_q(v^\pi_{\mathcal{U}^{sa}_p}) \qquad \text{(2d statement of Prop. 4.1)}$$

$$= R_0(s, a) + \gamma \sum_{s', a'} P_0(s'|s, a) \pi_{s'}(a') Q^\pi_{\mathcal{U}^{sa}_p}(s', a') - \alpha_{sa} - \gamma \beta_{sa} \kappa_q(v^\pi_{\mathcal{U}^{sa}_p})$$

$$= T^\pi_{(P_0, R_0)} Q^\pi_{\mathcal{U}^{sa}_p}(s, a) - \alpha_{sa} - \gamma \beta_{sa} \kappa_q(v^\pi_{\mathcal{U}^{sa}_p}).$$

$\square$

The above recursion applies the standard Bellman operator on robust Q-values. We can similarly apply it on the robust value function (itself can be computed efficiently based on [3, 14]).

**Corollary D.2.** *Let an $\ell_p$-ball constrained uncertainty set. Then, for all $(s, a) \in \mathcal{X}$ and $\pi \in \Pi$, the robust Q-value satisfies the following recursion in the $(s, a)$ and $s$-rectangular case respectively:*

$$Q^\pi_{\mathcal{U}^{sa}_p}(s, a) = R_0(s, a) + \gamma \sum_{s'} P_0(s'|s, a) v^\pi_{\mathcal{U}^{sa}_p}(s') - \alpha_{sa} - \gamma \beta_{sa} \kappa_q(v^\pi_{\mathcal{U}^s_p}),$$

$$Q^\pi_{\mathcal{U}^s_p}(s, a) = R_0(s, a) + \gamma \sum_{s'} P_0(s'|s, a) v^\pi_{\mathcal{U}^{sa}_p}(s') - \left( \frac{\pi_s(a)}{\|\pi_s\|_q} \right)^{q-1} \left( \alpha_s + \gamma \beta_s \kappa_q(v^\pi_{\mathcal{U}^s_p}) \right).$$

## D.2 Evaluation

Based on the Bellman recursion above, we now derive robust Q-learning equations to learn a robust Q-value. Precisely, we investigate if the linear operator below is contracting and can be evaluated efficiently:

$$(\mathcal{L}^\pi_{\mathcal{U}} Q)(s, a) := R^\pi_{\mathcal{U}, v}(s, a) + \gamma \sum_{(s', a') \in \mathcal{X}} P^\pi_{\mathcal{U}, v}(s'|s, a) \pi_{s'}(a') Q(s', a'), \quad \forall Q \in \mathbb{R}^{\mathcal{X}}, \qquad (9)$$

where $(P^\pi_{\mathcal{U}, v}, R^\pi_{\mathcal{U}, v}) \in \arg\min_{(P, R) \in \mathcal{U}} T^\pi_{(P, R)} v$ and $v(s) = \langle \pi_s, Q(s, \cdot) \rangle_{\mathcal{A}}, \quad \forall s \in \mathcal{S}$.

**Proposition D.3.** *Consider an $\ell_p$-ball constrained uncertainty set. Then, for all $Q \in \mathbb{R}^{\mathcal{X}}$ and $\pi \in \Pi$, the operator $\mathcal{L}^\pi$ can be evaluated as:*

$$(\mathcal{L}^\pi_{\mathcal{U}^{sa}_p} Q)(s, a) = T^\pi_{(P_0, R_0)} Q(s, a) - \alpha_{sa} - \gamma \beta_{sa} \kappa_q(v),$$

$$(\mathcal{L}^\pi_{\mathcal{U}^s_p} Q)(s, a) = T^\pi_{(P_0, R_0)} Q(s, a) - \left( \frac{\pi_s(a)}{\|\pi_s\|_q} \right)^{q-1} \left( \alpha_s + \gamma \beta_s \kappa_q(v) \right),$$

*where for all $Q \in \mathbb{R}^{\mathcal{X}}$, its corresponding value is $v(s) := \langle \pi_s, Q(s, \cdot) \rangle, \quad \forall s \in \mathcal{S}$.*

*Proof.* We give proof for the $(s,a)$-rectangular case only. The $s$-rectangular case follows the exact same lines except that we take the worst model for $s$-rectangular balls. By definition,

$$(\mathcal{L}_{\mathcal{U}_p^{\mathrm{sa}}}^\pi Q)(s,a) = \min_{(P,R)\in\mathcal{U}_p^{\mathrm{sa}}} \left\{ R(s,a) + \gamma \sum_{(s',a')\in\mathcal{X}} P(s'|s,a)\pi_{s'}(a')Q(s',a') \right\}$$

$$= \min_{R\in\mathcal{R}_p^{\mathrm{sa}}} R(s,a) + \gamma \min_{P\in\mathcal{P}_p^{\mathrm{sa}}} \left\{ \sum_{s'\in\mathcal{S}} P(s'|s,a)v(s') \right\}$$

$$= R_0(s,a) - \alpha_{s,a} + \gamma \sum_{s'\in\mathcal{S}} P_0(s'|s,a)v(s') - \beta_{s,a}\kappa_q(v) \qquad \text{(By [14])}$$

$$= R_0(s,a) - \alpha_{s,a} + \gamma \sum_{(s',a')\in\mathcal{X}} P_0(s'|s,a)\pi_{s'}(a')Q(s',a') - \beta_{s,a}\kappa_q(v)$$

$$= T_{(P_0,R_0)}^\pi Q(s,a) - \alpha_{s,a} - \beta_{s,a}\kappa_q(v).$$

$\square$

## D.3 Convergence

In the remainder of this section, we focus on $\ell_p$-ball constrained uncertainty sets of the form $\mathcal{U}_p^{\mathrm{sa}}$ or $\mathcal{U}_p^{\mathrm{s}}$. Let our Q-value iteration $Q_{n+1} := \mathcal{L}_{\mathcal{U}}^\pi Q_n$, and denote $v_n(s) = \langle \pi_s, Q_n(s,\cdot)\rangle_{\mathcal{A}}, \forall s \in \mathcal{S}, n \in \mathbb{N}$.

**Proposition D.4.** *For all $Q \in \mathbb{R}^{\mathcal{X}}$, denote $v(s) := \langle \pi_s, Q(s,\cdot)\rangle_{\mathcal{A}}, \forall s \in \mathcal{S}$. Then, for any policy $\pi \in \Pi$, the Q-value iteration defined according to $Q_{n+1} = \mathcal{L}_{\mathcal{U}}^\pi Q_n$ induces*

$$v_{n+1} := \mathcal{T}_{\mathcal{U}}^\pi v_n.$$

*Proof.* By construction, for all $s \in \mathcal{S}$ we have

$$v_{n+1}(s) = \langle \pi_s, Q_{n+1}(s,\cdot)\rangle_{\mathcal{A}}$$

$$= \langle \pi_s, (\mathcal{L}_{\mathcal{U}}^\pi Q_n)(s,\cdot)\rangle_{\mathcal{A}}$$

$$= \sum_{a\in\mathcal{A}} \pi_s(a)\left[ R_{\mathcal{U},v_n}^\pi(s,a) + \gamma \sum_{(s',a')\in\mathcal{X}} P_{\mathcal{U},v_n}^\pi(s'|s,a)\pi_{s'}(a')Q_n(s',a') \right] \qquad \text{(By Eq. 9)}$$

$$= \sum_{a\in\mathcal{A}} \pi_s(a)\left[ R_{\mathcal{U},v_n}^\pi(s,a) + \gamma \sum_{s'\in\mathcal{S}} P_{\mathcal{U},v_n}^\pi(s'|s,a)v_n(s') \right] \qquad \text{(By definition of } v_n)$$

$$= (\mathcal{T}_{\mathcal{U}}^\pi v_n)(s),$$

where the last equality holds because $(P_{\mathcal{U},v_n}^\pi, R_{\mathcal{U},v_n}^\pi) \in \arg\min_{(P,R)\in\mathcal{U}} \mathcal{T}_{(P,R)}^\pi v_n$. $\square$

As a result of the above proposition, the value iteration induced by our Q-value iteration rule converges linearly to the robust value function, i.e., $\|v_n - v_{\mathcal{U}}^\pi\|_\infty \le \gamma^n \|v_0\|_\infty$. Therefore, Q-value iterates converge to a fixed point. Precisely, $v_n \to_n v_{\mathcal{U}}^\pi$ implies that $(P_{\mathcal{U},v_n}^\pi, R_{\mathcal{U},v_n}^\pi) \to_n (P_{\mathcal{U}}^\pi, R_{\mathcal{U}}^\pi)$, which in turn implies that $Q_n \to_n Q_{\mathcal{U}}^\pi$. The result below further characterizes the convergence rate.

**Proposition D.5.** *For any policy $\pi \in \Pi$, the recursion $Q_{n+1} := \mathcal{L}_{\mathcal{U}}^\pi Q_n, n \in \mathbb{N}$ converges linearly to $Q_{\mathcal{U}}^\pi$.*

*Proof.* Thanks to Prop. D.4, we have:

$$\|Q_{n+1} - Q_{\mathcal{U}}^\pi\|_\infty = \|R_{\mathcal{U},v_n}^\pi + \gamma P_{\mathcal{U},v_n}^\pi v_n - R_{\mathcal{U}}^\pi + \gamma P_{\mathcal{U}}^\pi v_{\mathcal{U}}^\pi\|_\infty$$

$$= \gamma\|P_{\mathcal{U},v_n}^\pi v_n - P_{\mathcal{U}}^\pi v_{\mathcal{U}}^\pi\|_\infty \qquad (R_{\mathcal{U},v}^\pi = R_{\mathcal{U}}^\pi, \quad \forall v),$$

$$= \gamma\|(P_0 - B^\pi u_n)v_n - (P_0 - B^\pi u_{\mathcal{U}}^\pi)v_{\mathcal{U}}^\pi\|_\infty,$$

where by the worst kernel characterization, $B^\pi(s,a) := \beta_{s,a}$ for $\mathcal{U} = \mathcal{U}_p^{\mathrm{sa}}$ and $B^\pi(s,a) := \beta_s \left(\frac{\pi_s(a)}{\|\pi_s\|_q}\right)^{q-1}$ for $\mathcal{U} = \mathcal{U}_p^{\mathrm{sa}}$. This implies

$$
\begin{aligned}
\|Q_{n+1} - Q_\mathcal{U}^\pi\|_\infty &\leq \gamma\|P_0(v_n - v_\mathcal{U}^\pi)\|_\infty + \gamma\|B^\pi(u_n)^\top v_n - B^\pi(u_\mathcal{U}^\pi)^\top v_\mathcal{U}^\pi\|_\infty \\
&\leq \gamma^{n+1}\|v_0 - v_\mathcal{U}^\pi\|_\infty + \gamma\|(u_n)^\top v_n - (u_\mathcal{U}^\pi)^\top v_\mathcal{U}^\pi\| \qquad (B^\pi(s,a) \leq 1), \\
&\leq \gamma^{n+1}\|v_0 - v_\mathcal{U}^\pi\|_\infty + \gamma\|(u_n)^\top(v_n - v_\mathcal{U}^\pi)\| + \gamma\|(u_n - u_\mathcal{U}^\pi)^\top v_\mathcal{U}^\pi\| \\
&\leq \gamma^{n+1}\|v_0 - v_\mathcal{U}^\pi\|_\infty + \gamma^{n+1}S\|v_0 - v_\mathcal{U}^\pi\|_\infty + \gamma\frac{\|\mathcal{R}\|_\infty}{1-\gamma}\|u_n - u_\mathcal{U}^\pi\|_\infty.
\end{aligned}
$$

Here, $u_n, u_\mathcal{U}^\pi$ is the balanced-normalized vector associated with vector $v_n$ and $v_\mathcal{U}^\pi$, respectively. Recall that the $p$-balanced normalized vector $u$ associated with vector $v$ is given by

$$
u(s) := \frac{\mathrm{sign}(v(s) - \omega_q(v)\|v(s) - \omega_q(v)\|^{q-1}}{\kappa_q(v)^{q-1}}, \tag{10}
$$

where $\kappa_p(v) = \min_w\|v - w\mathbf{1}\|_p$ and $\omega_p(v) =_w \|v - w\mathbf{1}\|_p$. It is easy to see that $\omega_p, \kappa$ are Lipschitz in $v$. Hence, $\|u_n - u_\mathcal{U}^\pi\|_\infty \leq C \cdot \mathrm{Pol}(\|v_n - v_\mathcal{U}^\pi\|_\infty) \cdot \psi(\kappa(v_\mathcal{U}^\pi), S, A)$, where Pol is a polynomial and $\psi$ a real function. This implies that

$$
\|Q_{n+1} - Q_\mathcal{U}^\pi\|_\infty \leq \gamma^{n+1}\psi'(\kappa(v_\mathcal{U}^\pi), \|v_0 - v_\mathcal{U}^\pi\|_\infty, S, A),
$$

which concludes our proof. $\qquad\square$

# E  Robust Policy Gradient

**Theorem** (RPG). *For any rectangular $\ell_p$-ball-constrained uncertainty, the robust policy gradient is given by:*

$$
\partial_\pi \rho_\mathcal{U}^\pi = \sum_{(s,a)\in\mathcal{X}} \left(d_{P_0,\mu}^\pi(s) - c^\pi(s)\right) Q_\mathcal{U}^\pi(s,a)\nabla\pi_s(a), \tag{11}
$$

*where*

$$
c^\pi(s) := \frac{\gamma\langle d_{P_0,\mu}^\pi, \beta^\pi\rangle_\mathcal{S}}{1 + \gamma\langle d_{P_0,u_\mathcal{U}^\pi}^\pi, \beta^\pi\rangle_\mathcal{S}} d_{P_0,u_\mathcal{U}^\pi}^\pi(s), \quad \forall s \in \mathcal{S}.
$$

*Proof.* The proof directly follows from plugging the robust occupation measure of Thm. 4.6 and the robust Q-value into standard policy-gradient theorem [26]. $\qquad\square$

# F  Complexity Analysis

In this section, we aim to derive the complexity of one RPG iteration for different uncertainty sets. We first focus on non-robust MDPs, to then see how the complexity increases with the uncertainty structure.

**Non-robust MDPs.** Computing non-robust Q-values and occupation measure takes $O(S^2 A \log(\epsilon^{-1}))$ each, which represent the most expensive computations in PG. The product of $d^\pi, Q^\pi$ and $\nabla\pi$ in PG requires $O(SA)$ operations, which is insignificant. Hence, the total cost of PG corresponds to the computational cost of Q-values. More precisely, approximate the Q-value by $Q$ and the occupation measure by $d$ with $\frac{\epsilon}{SA}$ tolerance, that is, $\|Q - Q^\pi\|_\infty \leq \frac{\epsilon}{SA}$ and $\|d - d^\pi\|_\infty \leq \frac{\epsilon}{SA}$. This involves $O(S^2 A \log(SA\epsilon^{-1}))$ operations for each. Then, we have

$$
\sum_{(s,a)\in\mathcal{X}} d'(s)Q'(s,a)\nabla\pi_s(a) = \sum_{(s,a)\in\mathcal{X}} (Q^\pi(s,a) + \epsilon_1(s,a))(d^\pi(s,a) + \epsilon_2(s,a))\nabla\pi_s(a)
$$

where $\epsilon_1(s,a) := Q(s,a) - Q^\pi(s,a)$ and $\epsilon_2(s,a) := d(s,a) - d^\pi(s,a)$. Let $B$ be an upper bound of $\|Q^\pi\|_\infty$ and $\|d^\pi\|_\infty$. Then

$$
\begin{aligned}
\sum_{(s,a)\in\mathcal{X}} d'(s)Q'(s,a)\nabla\pi_s(a) &= \sum_{(s,a)\in\mathcal{X}} (Q^\pi(s,a) + \epsilon_1(s,a))(d^\pi(s,a) + \epsilon_2(s,a))\nabla\pi_s(a) \\
&= \sum_{(s,a)\in\mathcal{X}} Q^\pi(s,a)d^\pi(s,a)\nabla\pi_s(a) + O(\epsilon),
\end{aligned}
$$

so the exact complexity of policy gradient for non-robust MDP is $O(S^2 A \log(SA\epsilon^{-1}))$.

**Convex non-rectangular uncertainty set.** Robust policy improvement is strongly NP-Hard for non-rectangular uncertainty sets, even if convex [33]. The PG method finds global optimal given oracle access to policy gradient in polynomial time [30]. This implies that the computation of RPG must be NP-Hard.

### F.1  Helper results for Robust MDPs

**Variance and mean functions.** Computing $\kappa_p(v)$ (resp. $\omega_p(v)$) with $\epsilon$-tolerance requires $O(S \log(S\epsilon^{-1}))$ (resp. $O(S \log(\epsilon^{-1}))$) computations if we use binary search [14].

**Occupation measure.** Let $k \in \mathbb{R}^{\mathcal{S}}$ be any vector. By definition,

$$d^{\pi}_{P,k} = \sum_{n=0}^{\infty} \gamma^n k^{\top} (P^{\pi})^n$$

and

$$\left\| d^{\pi}_{P,k} - \sum_{n=0}^{N-1} \gamma^n k^{\top} (P^{\pi})^n \right\| = \left\| \sum_{n=N}^{\infty} \gamma^n k^{\top} (P^{\pi})^n \right\| \leq \|k\| \sum_{n=N}^{\infty} \|\gamma P^{\pi}\|^n.$$

Since $P^{\pi}$ is a stochastic matrix, $\|P^{\pi}\| \leq 1$ and

$$\left\| d^{\pi}_{P,k} - \sum_{n=0}^{N-1} \gamma^n k^{\top} (P^{\pi})^n \right\| \leq \frac{\|k\|\gamma^N \|P^{\pi}\|^N}{1 - \gamma\|P^{\pi}\|} \leq \frac{\|k\|\gamma^N}{1 - \gamma}.$$

This implies that $\sum_{n=0}^{N-1} \gamma^n k^{\top} (P^{\pi})^n$ is an $O(\gamma^N)$ approximation of $d^{\pi}_{P,k}$. Now, take $u_0 = k$ and

$$u_{n+1} := \gamma(u_n)^{\top} P,$$

then $\sum_{n=0}^{N-1} \gamma^n k^{\top} (P^{\pi})^n = \sum_{n=0}^{N-1} u_n$. Each iteration takes $O(S^2)$ computations, leading to total cost $O(S^2 N)$ for $N$ iterations. Computing $P^{\pi}$ from $P$ is $O(S^2 A)$. We conclude that computing the occupation measure has a complexity of $O(S^2 A + S^2 \log(\epsilon^{-1}))$.

**Lemma F.1.** *We can approximate $d^{\pi}_{P,k}$ by $\sum_{n=0}^{N-1} \gamma^n (k')^{\top} (P^{\pi})^n$ with complexity $O(S^2 A + S^2 \log(\epsilon^{-1}))$ and tolerance*

$$\left\| d^{\pi}_{P,k} - \sum_{n=0}^{N-1} \gamma^n k'^{\top} (P^{\pi})^n \right\| \leq O\left( \frac{\|k\|\gamma^N + \|k - k'\|}{1 - \gamma} \right).$$

*Proof.*

$$\left\| d^{\pi}_{P,k} - \sum_{n=0}^{N-1} \gamma^n k'^{\top} (P^{\pi})^n \right\| \leq \left\| d^{\pi}_{P,k} - \sum_{n=0}^{N-1} \gamma^n k^{\top} (P^{\pi})^n \right\| + \left\| \sum_{n=0}^{N-1} \gamma^n k'^{\top} (P^{\pi})^n - \sum_{n=0}^{N-1} \gamma^n k^{\top} (P^{\pi})^n \right\|$$

$$\leq O\left( \frac{\|k\|\gamma^N}{1 - \gamma} \right) + \left\| \sum_{n=0}^{N-1} \gamma^n k^{\top} (P^{\pi})^n - \sum_{n=0}^{N-1} \gamma^n k'^{\top} (P^{\pi})^n \right\|$$

$$\leq O\left( \frac{\|k\|\gamma^N}{1 - \gamma} \right) + \|k - k'\| \left\| \sum_{n=0}^{N-1} \gamma^n (P^{\pi})^n - \sum_{n=0}^{N-1} \gamma^n (P^{\pi})^n \right\|$$

$$\leq O\left( \frac{\|k\|\gamma^N}{1 - \gamma} \right) + O\left( \frac{\|k - k'\|}{1 - \gamma} \right).$$

$\square$

**Computing Q-value given value function.** Let $v$ be an $\epsilon_1$ approximation of robust value function $v_{\mathcal{U}}^\pi$, that is $\|v - v_{\mathcal{U}}^\pi\|_\infty \le \epsilon_1$. We want to compute the Q-value using the relation:

$$Q_{\mathcal{U}}^\pi(s,a) = R_{\mathcal{U}}^\pi(s,a) + \sum_{s,a} \pi_s(a) P_{\mathcal{U}}^\pi(s'|s,a) v_{\mathcal{U}}^\pi(s')$$

$$= R_0(s,a) + \gamma \sum_{s'} P_0(s'|s,a) v_{\mathcal{U}}^\pi(s') - \Omega_{\mathcal{U}}(v_{\mathcal{U}}^\pi, \pi).$$

where $\Omega_{\mathcal{U}_p^{\mathrm{s}}}(v_{\mathcal{U}_p^{\mathrm{s}}}^\pi, \pi) = \frac{\pi_s(a)^{q-1}}{\|\pi_s\|_q^{q-1}} \left( \alpha_s + \gamma\beta_s\kappa_q(v_{\mathcal{U}_p^{\mathrm{s}}}^\pi) \right)$ and $\Omega_{\mathcal{U}_p^{\mathrm{sa}}}(v_{\mathcal{U}_p^{\mathrm{sa}}}^\pi, \pi) = \alpha_{sa} + \gamma\beta_{sa}\kappa_q(v_{\mathcal{U}_p^{\mathrm{sa}}}^\pi)$. Let $Q$ be approximated from $v$ by

$$Q(s,a) = R_0(s,a) + \gamma \sum_{s'} P_0(s'|s,a) v(s') - \Omega_{\mathcal{U}}(v, \pi).$$

We thus have

$$
\begin{aligned}
\|Q_{\mathcal{U}}^\pi(s,a) - Q(s,a)\|_\infty &= \gamma \left\| \sum_{s'\in\mathcal{S}} P_0(s'|s,a)(v(s') - v_{\mathcal{U}}^\pi) \right\| + \|\Omega_{\mathcal{U}}(v,\pi) - \Omega_{\mathcal{U}}(v_{\mathcal{U}}^\pi,\pi)\| \\
&\le \gamma\epsilon_1 + \|\Omega_{\mathcal{U}}(v,\pi) - \Omega_{\mathcal{U}}(v_{\mathcal{U}}^\pi,\pi)\| \\
&\le \gamma\epsilon_1 + \|\beta\|_\infty \|\kappa_q(v) - \kappa_q(v_{\mathcal{U}}^\pi)\| \\
&\le \gamma\epsilon_1 + \|\beta\|_\infty S^{\frac{1}{q}}\epsilon_1 \qquad\qquad\qquad \text{(By Lemma F.2)} \\
&= O(S^{\frac{1}{q}}\epsilon_1),
\end{aligned}
$$

so that $\|Q - Q_{\mathcal{U}}^\pi\|_\infty \le O(S^{\frac{1}{q}}\epsilon_1)$.

**Lemma F.2.** *The variance function $\kappa_p$ is Lipschitz. More precisely,*

$$\|\kappa_p(v_1) - \kappa_p(v_2)\| \le S^{\frac{1}{p}}\|v_1 - v_2\|_\infty \le S^{\frac{1}{p}}\|v_1 - v_2\|_\infty, \quad \forall v_1, v_2 \in \mathbb{R}^{\mathcal{S}}.$$

*Proof.* Let $v_i \in \mathbb{R}^{\mathcal{S}}$ and $w_i \in \arg\min_{w\in\mathbb{R}}\|v_i - w\mathbf{1}\|_p$ for $i = 1,2$. Without loss of generality, further assume that $\kappa_p(v_1) \ge \kappa_p(v_2)$. Then,

$$
\begin{aligned}
\|\kappa_p(v_1) - \kappa_p(v_2)\| &= \kappa_p(v_1) - \kappa_p(v_2) \\
&= \min_{w\in\mathbb{R}}\|v_1 - w\mathbf{1}\|_p - \min_{w\in\mathbb{R}}\|v_2 - w\mathbf{1}\|_p & \text{(By definition)} \\
&\le \|v_1 - w_2\mathbf{1}\|_p - \|v_2 - w_2\mathbf{1}\|_p & \text{(By definition of } \omega_2\text{)} \\
&\le \|(v_1 - w_2\mathbf{1}) - (v_2 - w_2\mathbf{1})\|_p & \text{(Reverse triangle inequality)} \\
&= \|v_1 - v_2\|_p \\
&= \left( \sum_{s\in\mathcal{S}} (v_1(s) - v_2(s))^p \right)^{\frac{1}{p}} \le S^{\frac{1}{p}}\|v_1 - v_2\|_\infty.
\end{aligned}
$$

$\square$

**Lemma F.3.** $Q_{\mathcal{U}_p^{\mathrm{sa}}}^\pi$ *can be approximated to $\epsilon$ tolerance with the same complexity as computing $v_{\mathcal{U}_p^{\mathrm{sa}}}^\pi$ to $S^{-\frac{1}{q}}\epsilon$.*

*Proof.* We can compute the value function with $S^{-\frac{1}{q}}\epsilon$ tolerance. The rest of the operations are insignificant. The result follows from above. $\square$

### F.2 RPG Complexity

Let $O_p^{\mathrm{sa}}(\epsilon)$ be the complexity of computing $v_{\mathcal{U}_p^{\mathrm{sa}}}^\pi$ with $\epsilon$-tolerance (see [14] for more details). Calculating Q-value up to $\epsilon_1$ requires $O_p^{\mathrm{sa}}(S^{-\frac{1}{q}}\epsilon_1)$ computations according to Lemma F.3. Letting $d_1$ and $d_2$ be $\epsilon_2$-approximations of $d_{P_0,\mu}^\pi$ and $d_{P_0,k}^\pi$ respectively, their complexity is insignificant compared to $O_p^{\mathrm{sa}}(S^{-\frac{1}{q}}\epsilon_2)$. Now, approximating the gradient using $d_1, d_2, Q, \nabla\pi$ as in Thm. 5.1 has a complexity

of $O(SA)$. Since the uncertainty set $\mathcal{U}$ is compact, all quantities are bounded. There are $O(SA)$ operations in Thm. 5.1, so taking $\epsilon_1, \epsilon_2 = O(\frac{\epsilon}{SA})$, we get $O(\epsilon)$ for the gradient. Hence, the total complexity is $O_p^{\text{sa}}(S^{-\frac{1}{q}-1}A^{-1}\epsilon)$ which is $\tilde{O}(S^2 A \log(\epsilon^{-1}))$ by hiding log factors. A similar analysis follows for the $s$-rectangular case.

# G   Generalization to arbitrary norms

In this section, we analyze how to generalize our result to general norms that are not $\ell_p$.

## G.1   $(s, a)$-rectangular robust MDPs

Consider an $(s, a)$-rectangular uncertainty set $\mathcal{U} = \mathcal{U}_{\|\cdot\|}^{\text{sa}}$ constrained by:

$$\mathcal{U}_{\|\cdot\|}^{\text{sa}} = (P_0 + \mathcal{P}) \times (R_0 + \mathcal{R}), \qquad \text{where} \qquad (\mathcal{P}, \mathcal{R}) = (\times_{s,a}\mathcal{P}_{sa}, \times_{s,a}\mathcal{P}_{sa}),$$

$$\mathcal{R}_{(s,a)} = \{r \in \mathbb{R} \mid \|r\| \le \alpha_{s,a}\}, \quad \text{and} \quad \mathcal{P}_{(s,a)} = \{p \in \mathbb{R}^{\mathcal{S}} \mid \langle p, \mathbf{1} \rangle_{\mathcal{S}} = 0, \|p\| \le \beta_{s,a}\}.$$

The robust Bellman operator $\mathcal{T}_{\mathcal{U}}^{\pi}$ can be evaluated as

$$(\mathcal{T}_{\mathcal{U}}^{\pi}v)(s) = \sum_a \pi_s(a) \left[ R(s, a) - \gamma\beta_{s,a}\kappa_{\|\cdot\|}(v) + \gamma\sum_{s'} P(s'|s, a)v(s') \right],$$

where the variance function is defined as

$$\kappa_{\|\cdot\|}(v) := \min_{\langle u, \mathbf{1} \rangle_{\mathcal{S}}=0, \|u\| \le 1} \langle u, v_{\mathcal{U}}^{\pi} \rangle.$$

This can be used to compute the robust value function. Then the worst values can found using robust Bellman operator $\mathcal{T}_{\mathcal{U}}^{\pi}$ and robust value function $v_{\mathcal{U}}^{\pi}$ as

$$(P_{\mathcal{U}}^{\pi}, R_{\mathcal{U}}^{\pi}) \in \underset{(P,R)\in\mathcal{U}}{\arg\min} \, \mathcal{T}_{(P,R)}^{\pi}v_{\mathcal{U}}^{\pi}, \qquad [33].$$

It is easy to see that the worst values are given as

$$R_{\mathcal{U}}^{\pi}(s, a) = R_0(s, a) - \alpha_{s,a} \quad \text{and} \quad P_{\mathcal{U}}^{\pi}(\cdot|s, a) = P_0^{\pi}(\cdot|s, a) - \beta_{s,a}u_{\mathcal{U}}^{\pi},$$

where normalized-balanced value function $u_{\mathcal{U}}^{\pi}$ is a solution to

$$\min_{\langle u, \mathbf{1} \rangle_{\mathcal{S}}=0, \|u\| \le 1} \langle u, v_{\mathcal{U}}^{\pi} \rangle.$$

Observe that the worst kernel is still a rank-one perturbation of the nominal kernel. Hence, the robust occupation measure can be obtained using Lemma 4.4 as

$$d_{\mathcal{U},\mu}^{\pi} = d_{P_0,\mu}^{\pi} - \gamma\frac{\langle d_{P_0,\mu}^{\pi}, \beta^{\pi} \rangle_{\mathcal{S}}}{1 + \gamma\langle d_{P_0,u_{\mathcal{U}}^{\pi}}^{\pi}, \beta^{\pi} \rangle_{\mathcal{S}}} d_{P_0,u_{\mathcal{U}}^{\pi}}^{\pi}, \tag{12}$$

where $\beta^{\pi}(s) = \sum_a \pi_s(a)\beta_{s,a}$. The last ingredient to compute RPG is the robust Q-value which can be computed using robust value function and worst values. However, it can be computed directly using the following iterates:

$$Q_{n+1}(s) = \min_{(P,R)\in\mathcal{U}} \left[ R(s, a) + \gamma\sum_{s'} P(s'|s, a)v(s') \right]$$

$$= R(s, a) - \alpha_{s,a} - \gamma\beta_{s,a}\kappa_{\|\cdot\|}(v) + \gamma\sum_{s'} P(s'|s, a)v(s'),$$

as $Q_n$ converges to robust Q-value $Q_{\mathcal{U}}^{\pi}$ linearly.

The proofs of the above claims are very similar to the $\ell_p$ case so they are omitted. Finally, computing the variance function $\kappa_{\|\cdot\|}$ and the normalized-value function $u_{\mathcal{U}}^{\pi}$ can be done using numerical convex optimization methods for general norms. For the $\ell_p$ case, these can be obtained in concrete form, so we choose to focus on $\ell_p$ in our main study.

## G.2  $s$-rectangular case

Generalization to $s$-rectangular balls of a general norm is not straightforward and may not be possible for all types of norms. The crucial property of the $\ell_p$-norm exploited in our rank-one perturbation proof is 'decoupling', that is, for all $x \in \mathbf{R}^{\mathcal{A} \times \mathcal{S}}$, there must exist $k, l, m \in \mathbb{R}_+$ such that

$$\|x\|_p^k = \sum_{a \in \mathcal{A}} \|x_a\|_m^l.$$

This holds in the $\ell_p$ case with $k = l = m = p$. Further analysis of this setting is left for future work.

## G.3  Generalization to metrics and divergences

Generalizing RPG to distance or divergence-based uncertainty sets is also not obvious. Our RPG method, in particular the robust occupation measure, crucially relies on the rank-one perturbation characterization of the worst kernel, which might not apply for example with KL-ball constraints. We leave this analysis for future work.

# H  Experiments

**Parameters.** All the nominal transition kernels and reward functions are generated randomly. The number of states and the number of actions are varied. Discount factor $\gamma = 0.9$, reward noise radius $\alpha_{s,a}, \alpha_s = 0.1$, transition noise kernel $\beta_{s,a}, \beta_s = \frac{0.01}{SA}$.

**Hardware** Experiments are done on the machine with the following configuration: Intel(R) Core(TM) i7-6700 CPU @3.40GHZ, size:3598MHz, capacity 4GHz, width 64 bits, memory size 64 GiB.

**Software and codes** All the experiments were done in Python using numpy, matplotlib. All codes and results are available at https://github.com/navdtech/rpg.

**Procedure and Results.** All the experiments were repeated 100 times, except for Linear Programming (LP) cases as LP methods were very time-consuming. In LP methods, experiments were repeated 5 times except for the case ($S = 500, A = 100$) which was done only once. As this case was prohibitively expensive. Standard deviation in all cases was less than $10\%$, and typically $1 - 2\%$. This conveniently illustrates the superiority of our methods over LP methods.

**Observations**

- **Scalability of our methods.** Note that our methods scale very well with large state-action space. It takes a (small) constant times the time required by non-robust MDPs. On the other hand, LP methods explode. Both observations confirm the theoretical time complexity.

- **sa-case vs s case in LP methods.** We see s-case outperforms sa-case for small state-action spaces via LP methods. This is opposite to the theoretical time complexity of s-case which expensive than sa-case. We believe this is due to the internal implementation issues. Note that computing the robust value function is the most expensive step which requires evaluation of the robust Bellman operator. In sa case, one evaluation requires solving $SA$ LP programs with $S$ variables each, while for s-case, it is $S$ LP programs with $SA$ variables each. To solve LP, scipy.linprog is used, we believe it does some parallelization for large LPs. Hence, we observe less cost for sa-case. However, we observe that the cost of s-case increases much faster than s-case, and eventually under-performing than sa-case.

## H.1  RPG by LP

We compute RPG using LP in the following steps:

1. **(Robust Value Iteration)** Approximately compute the robust value function $v_{\mathcal{U}}^\pi$ using the iterates $v_{n+1} := \mathcal{T}_{\mathcal{U}}^\pi v_n$. Evaluation of robust Bellman operator $\mathcal{T}_{\mathcal{U}}^\pi$ is done via LPs as described below. This is the most expensive step as it requires evaluating robust Bellman operators $O(\log(\epsilon^{-1}))$ times, and each evaluation requires many LPs.

2. **(Adversarial Values)** Compute the worst values $(P_{\mathcal{U}}^{\pi}, R_{\mathcal{U}}^{\pi})$ using the robust value function from the following relation:

$$(P_{\mathcal{U}}^{\pi}, R_{\mathcal{U}}^{\pi}) \in \underset{(P,R) \in \mathcal{U}}{\arg\min} \, \mathcal{T}_{\mathcal{U}}^{\pi} v_{\mathcal{U}}^{\pi}.$$

This is also solved by LP.

3. **(Policy Gradient Theorem)** We now compute the RPG using policy gradient Theorem [26] w.r.t. the adversarial values computed above, as

$$\partial \rho_{\mathcal{U}}^{\pi} = \sum_{s,a} d_{P_{\mathcal{U}}^{\pi}}^{\pi}(s) Q_{P_{\mathcal{U}}^{\pi}, R_{\mathcal{U}}^{\pi}}^{\pi}(s, a) \nabla \pi_s(a).$$

Observe that $d_P^{\pi}$ can be approximated as $\sum_{n=0}^{n} (\gamma P^{\pi})^n$ for large $n$ enough, and $Q_{P,R}^{\pi}$ can be approximated by dynamic programming [25]. Notably, this step and the second step are negligible as compared to the first step.

**Robust Value Iteration by LP**

`sa`**-rectangular robust MDPs.** We first consider `sa`-rectangular $L_1$ constrained uncertainty set $\mathcal{U}_p^{\text{sa}} = \mathcal{P} \times \mathcal{R}$. Robust Bellman operator is given by

$$(\mathcal{T}_{\mathcal{U}_p^{\text{sa}}}^{\pi} v)(s) = \max_a \underbrace{\min_{p \in \mathcal{P}_{sa}, r \in \mathcal{R}_{sa}} \left[ r + \gamma \sum_{s'} p(s') v(s') \right]}_{\text{LP with } S \text{ variable}}.$$

Note that the above can be solved by $A$ LPs as uncertainty set $\mathcal{U}_p^{\text{sa}} = \mathcal{P} \times \mathcal{R}$ induces linear constraint and the objective is also linear with $S$ variables. Hence, evaluation of $\mathcal{T}_{\mathcal{U}_p^{\text{sa}}}^{\pi} v$ requires solving $SA$ LPs with $S$ variable each.

$s$**-rectangular robust MDPs.** We now consider $s$-rectangular $L_1$ constrained uncertainty set $\mathcal{U}_p^{\text{s}} = \mathcal{P} \times \mathcal{R}$. Robust Bellman operator is given by

$$(\mathcal{T}_{\mathcal{U}_p^{\text{s}}}^{\pi} v)(s) = \underbrace{\min_{p \in \mathcal{P}_s, r \in \mathcal{R}_s} \sum_a \pi_s(a) \left[ r(a) + \gamma \sum_{s'} p(s'|a) v(s') \right]}_{\text{LP with } SA \text{ variable}}.$$

Note that the above can be solved by one LP as uncertainty set $\mathcal{U}_p^{\text{s}} = \mathcal{P} \times \mathcal{R}$ induces linear constraint and the objective is also linear with $SA$ variables. Hence, evaluation of $\mathcal{T}_{\mathcal{U}_p^{\text{s}}}^{\pi} v$ requires solving $S$ LPs with $SA$ variable each.

