# Policy Gradient for Rectangular Robust Markov Decision Processes

## Abstract

Policy gradient methods have become a standard for training reinforcement learning agents in a scalable and efficient manner. However, they do not account for transition uncertainty, whereas learning robust policies can be computationally expensive. In this paper, we introduce robust policy gradient (RPG), a policy-based method that efficiently solves rectangular robust Markov decision processes (MDPs). We provide a closed-form expression for the worst occupation measure. Incidentally, we find that the worst kernel is a rank-one perturbation of the nominal. Combining the worst occupation measure with a robust Q-value estimation yields an explicit form of the robust gradient. Our resulting RPG can be estimated from data with the same time complexity as its non-robust equivalent. Hence, it relieves the computational burden of convex optimization problems required for training robust policies by current policy gradient approaches.

## 1 Introduction

Markov decision processes (MDPs) provide an analytical framework to solve sequential decision-making problems and seek the best performance in a fixed environment. Since the resulting policy can be highly sensitive to parameter values [16], the robust MDP setting alternatively maximizes return under the worst scenario, thus yielding robustness to uncertain environments [18, 10]. In practice, the robust MDP paradigm quantifies the level of uncertainty through a set $\mathcal{U}$ determining the possible range of model perturbations. Then, a policy is said to be robust-optimal if it reaches maximal performance under the most adversarial model within the uncertainty set. Developing efficient solvers for robust MDPs is of great interest, as it can lead to behavior policies with generalization guarantees [31].

If not computationally expensive, robust MDPs can be strongly NP-hard [30]. Thus, to preserve tractability, we commonly assume that $\mathcal{U}$ is convex and $s$-rectangular, i.e., $\mathcal{U} = \times_{s \in \mathcal{S}} \mathcal{U}_s$ [18, 10, 30]. The latter assumption means that the overall uncertainty should be designed independently for each state. Further simplification may consider $(s, a)$-rectangular uncertainty sets of the form $\mathcal{U} = \times_{(s,a) \in \mathcal{X}} \mathcal{U}_{(s,a)}$, albeit this naturally leads to more conservative strategies. In any case, planning in robust MDPs can be computationally costly, as it involves successive max-min problems [7, 1, 30]. To address this issue, the works [3, 12] have established an equivalence between robustness and regularization in reinforcement learning (RL) in order to derive efficient robust planning methods for $s$ and $(s, a)$-rectangular robust MDPs. Indeed, it appears that resorting to proper regularization instead of solving a minimization problem can yield robust behavior without requiring the polynomial time complexity of convex optimization problems [3].

Alternatively to planning, policy gradient algorithms (PG) directly learn an optimal policy by applying gradient steps towards better performance [22]. Due to its scalability, ease of implementation, and adaptability to many different settings such as model-free and continuous state-action spaces [11, 21],

PG has become the workhorse of RL. Although regularization techniques such as max-entropy [6] or Tsallis [13] have shown robust behavior without impairing computational cost, they only account for adversarial reward [2, 5, 3]. Differently, robust PG formulations (RPG) formulations aim to address uncertainty to reward *and* transition functions.

Despite their ability to propel robust behavior, RPG methods that target robust optimal policies are still rare in the RL literature. The global convergence of RPG established in [14, 27] further motivates us to come up with a practical method for estimating the gradients. In fact, [14]

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

 noise radius $\beta$ should be small enough so that transition kernels of the form $P_0 + \mathcal{P}$ are well defined. This normed ball structure on the uncertainty sets enables us to compute robust Bellman updates with similar time complexity as non-robust ones using regularization [3, 12].

First, define the generalized variance function and the mean function as

$$\kappa_q(v) = \min_{w \in \mathbb{R}} \|v - w\mathbf{1}\|_q, \qquad \omega_q(v) \in \arg\min_{w \in \mathbb{R}} \|v - w\mathbf{1}\|_q,$$

respectively, where $q$ is the conjugate value of $p$ (see Tab. 2 for their closed-form expression when $q \in \{1, 2, \infty\}$). Then, we can efficiently evaluate robust value functions by regularizing a standard Bellman operator instead of solving a minimization.

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

}$ for uncertainty set $\mathcal{U} = \mathcal{U}_p^{\mathrm{sa}}, \mathcal{U}_p^{\mathrm{s}}$ for every policy $\pi \in \Pi$.*

*Proof.*

$$
\begin{aligned}
\|Q_{n+1} - Q_{\mathcal{U}}^{\pi}\|_{\infty} &= \|R_{\mathcal{U},v_n}^{\pi} + \gamma P_{\mathcal{U},v_n}^{\pi} v_n - R_{\mathcal{U}}^{\pi} + \gamma P_{\mathcal{U}}^{\pi} v_{\mathcal{U}}^{\pi}\|_{\infty}, \\
&= \gamma \|P_{\mathcal{U},v_n}^{\pi} v_n - P_{\mathcal{U}}^{\pi} v_{\mathcal{U}}^{\pi}\|_{\infty}, \qquad (\text{ as } R_{\mathcal{U},v}^{\pi} = R_{\mathcal{U}}^{\pi}, \quad \forall v), \\
&= \gamma \|(P_0 - B^{\pi} u_n)v_n - (P_0 - B^{\pi} u_{\mathcal{U}}^{\pi})v_{\mathcal{U}}^{\pi}\|_{\infty}, \qquad (\text{ as } R_{\mathcal{U},v}^{\pi} = R_{\mathcal{U}}^{\pi}, \quad \forall v),
\end{aligned}
$$

where $B^{\pi}(s,a) = \beta_{s,a}$ for $\mathcal{U} = \mathcal{U}_p^{\mathrm{sa}}$ and $B^{\pi}(s,a) = \beta_s \left( \frac{\pi_s(a)}{\|\pi_s\|_q} \right)^{q-1}$ for $\mathcal{U} = \mathcal{U}_p^{\mathrm{sa}}$. The equality comes from the worst kernel characterization. This implies

$$\|Q_{n+1} - Q_{\mathcal{U}}^\pi\|_\infty \leq \gamma\|P_0(v_n - v_{\mathcal{U}}^\pi)\|_\infty + \gamma\|B^\pi(u_n)^\top v_n - B^\pi(u_{\mathcal{U}}^\pi)^\top v_{\mathcal{U}}^\pi\|_\infty, \qquad ,$$
$$\leq \gamma^{n+1}\|v_0 - v_{\mathcal{U}}^\pi\|_\infty + \gamma\|(u_n)^\top v_n - (u_{\mathcal{U}}^\pi)^\top v_{\mathcal{U}}^\pi\|, \qquad (\text{as } B^\pi(s,a) \leq 1),$$
$$\leq \gamma^{n+1}\|v_0 - v_{\mathcal{U}}^\pi\|_\infty + \gamma\|(u_n)^\top(v_n - v_{\mathcal{U}}^\pi)\| + \gamma\|(u_n - u_{\mathcal{U}}^\pi)^\top v_{\mathcal{U}}^\pi\|,$$
$$\leq \gamma^{n+1}\|v_0 - v_{\mathcal{U}}^\pi\|_\infty + \gamma^{n+1}S\|v_0 - v_{\mathcal{U}}^\pi\|_\infty + \gamma\frac{\|\mathcal{R}\|_\infty}{1-\gamma}\|u_n - u_{\mathcal{U}}^\pi\|_\infty.$$

Here, $u_n, u_{\mathcal{U}}^\pi$ is the balanced normalized vector associated with vector $v_n$ and $v_{\mathcal{U}}^\pi$ respectively. Recall, the $p$-balanced normalized vector $u$ associated with vector $v$ is given by

$$u(s) := \frac{\text{sign}(v(s) - \omega_q(v)\|v(s) - \omega_q(v)\|^{q-1}}{\kappa_q(v)^{q-1}}, \tag{10}$$

where $\kappa_p(v) = \min_w \|v - w\mathbf{1}\|_p$ and $\omega_p(v) =_w \|v - w\mathbf{1}\|_p$. It is easy to see that $\omega_p, \kappa$ are Lipschitz function in $v$. Hence, $\|u_n - u_{\mathcal{U}}^\pi\|_\infty \leq CPol(\|v_n - v_{\mathcal{U}}^\pi\|_\infty)f(\kappa(v_{\mathcal{U}}^\pi), S, A)$, where $Pol$, $f$ is some polynomial and some function. This implies,

$$\|Q_{n+1} - Q_{\mathcal{U}}^\pi\|_\infty \leq \gamma^{n+1}f(\kappa(v_{\mathcal{U}}^\pi), \|v_0 - v_{\mathcal{U}}^\pi\|_\infty, S, A).$$

This concludes our proof. $\qquad\qquad\square$

## E   Robust Policy Gradient

**Theorem** (RPG). *For any rectangular $\ell_p$-ball-constrained uncertainty, the robust policy gradient is given by:*

$$\partial_\pi \rho_{\mathcal{U}}^\pi = \sum_{(s,a)\in\mathcal{X}} \left(d_{P_0,\mu}^\pi(s) - c^\pi(s)\right) Q_{\mathcal{U}}^\pi(s,a)\nabla\pi_s(a), \tag{11}$$

*where*

$$c^\pi(s) := \frac{\gamma\langle d_{P_0,\mu}^\pi, \beta^\pi\rangle_\mathcal{S}}{1 + \gamma\langle d_{P_0,u_{\mathcal{U}}^\pi}^\pi, \beta^\pi\rangle_\mathcal{S}} d_{P_0,u_{\mathcal{U}}^\pi}^\pi(s), \quad \forall s \in \mathcal{S}.$$

*Proof.* The proof directly follows from plugging the robust occupation measure of Thm. 4.6 and the robust Q-value into standard policy-gradient theorem [23]. $\qquad\square$

## F   Complexity Analysis

In this section, we study the iteration complexity to compute robust policy gradient different uncertainty set.

**Convex non-rectangular Uncertainty set.** Robust policy improvement are strongly NP Hard for non-rectangular uncertainty set, even if it is convex [30]. The policy gradient method finds global optimal given oracle access to policy gradient in polynomial time [27]. This implies computation of policy gradient must be of NP Hard.

**Non-robust MDPs.** For non-robust case, computation of Q-value and occupation is $O(S^2 A \log(\epsilon^{-1}))$ each, which are most costly computations. Computing the product of $d^\pi, Q^\pi$ and $\nabla\pi$ as in policy gradient is $O(SA)$ operation, which insignificant. Hence, the total cost for computing policy gradient is the same as cost of Q-value. More precisely, lets approximate Q-value with $Q$ and occupation with $d$, with $\frac{\epsilon}{SA}$ tolerance, that is $\|Q - Q^\pi\|_\infty, \|d - d^\pi\|_\infty \leq \frac{\epsilon}{SA}$. This takes $O(S^2 A \log(SA\epsilon^{-1}))$ each. Now then, we have

$$\sum_{s,a} d'(s)Q'(s,a)\nabla\pi_s(a) = \sum_{s,a}(Q^\pi(s,a) + \epsilon_1(s,a))(d^\pi(s,a) + \epsilon_2(s,a))\nabla\pi_s(a)$$

where $\epsilon_1(s,a) = Q(s,a) - Q^\pi(s,a)$ and $\epsilon_2(s,a) = d(s,a) - d^\pi(s,a)$. We know, let $B$ be the bound on $\|Q^\pi\|_\infty, \|d^\pi\|_\infty \leq B$. So now we have,

$$\sum_{s,a} d'(s)Q'(s,a)\nabla \pi_s(a) = \sum_{s,a} (Q^\pi(s,a) + \epsilon_1(s,a))(d^\pi(s,a) + \epsilon_2(s,a))\nabla \pi_s(a)$$
$$= \sum_{s,a} Q^\pi(s,a)d^\pi(s,a)\nabla \pi_s(a) + O(\epsilon).$$

564 So the exact complexity of policy gradient for non-robust MDP is $O(S^2 A \log(SA\epsilon^{-1}))$.

565 ## F.1   Helper results for Robust MDPs

566 **Computing variance and mean functions.** Computing $\kappa_p(v)$ and $\omega_p(v)$ to $\epsilon$ tolerance requires
567 $O(S \log(S\epsilon^{-1}))$ and $O(S \log(\epsilon^{-1}))$ respectively, via binary search [12].

568 **Computing occupation measure.** Let $k \in \mathbb{R}^S$ be any vector. From definition, we have

$$d^\pi_{P,k} = \sum_{n=0}^{\infty} \gamma^n k^\top (P^\pi)^n.$$

569 We have,

$$\|d^\pi_{P,k} - \sum_{n=0}^{N-1} \gamma^n k^\top (P^\pi)^n\| = \|\sum_{n=N}^{\infty} \gamma^n k^\top (P^\pi)^n\| \leq \|k\| \sum_{n=N}^{\infty} \|\gamma P^\pi\|^n.$$

570 Since, $\|P^\pi\| \leq 1$ as it is a stochastic matrix, then

$$\|d^\pi_{P,k} - \sum_{n=0}^{N-1} k^\top \gamma^n (P^\pi)^n\| \leq \frac{\|k\|\gamma^N \|P^\pi\|^N}{1 - \gamma\|P^\pi\|} \leq \frac{\|k\|\gamma^N}{1 - \gamma}.$$

571 This implies $\sum_{n=0}^{N-1} \gamma^n k^\top (P^\pi)^n$ is $O(\gamma^N)$ approximation of $d^\pi_{P,k}$. Now, take $u_0 = k$ and

$$u_{n+1} := \gamma(u_n)^\top P,$$

572 then $\sum_{n=0}^{N-1} \gamma^n k^\top (P^\pi)^n = \sum_{n=0}^{N-1} u_n$. And each iteration take $O(S^2)$ iterations, leading to total
573 cost $O(S^2 N)$ for $N$ iterations. Computing $P^\pi$ from $P$ is $O(S^2 A)$. We conclude computing the
574 occupation measure has complexity of $O(S^2 A + S^2 \log(\epsilon^{-1}))$.

575 **Lemma F.1.** *We can approximate $d^\pi_{P,k}$ with $\sum_{n=0}^{N-1} \gamma^n (k')^\top (P^\pi)^n$ with complexity $O(S^2 A +$*
576 *$S^2 \log(\epsilon^{-1}))$ with tolerance:*

$$\|d^\pi_{P,k} - \sum_{n=0}^{N-1} \gamma^n (k')^\top (P^\pi)^n\| \leq O(\frac{\|k\|\gamma^N + \|k - k'\|}{1 - \gamma})$$

*Proof.*

$$\|d^\pi_{P,k} - \sum_{n=0}^{N-1} \gamma^n (k')^\top (P^\pi)^n\| \leq \|d^\pi_{P,k} - \sum_{n=0}^{N-1} \gamma^n (k)^\top (P^\pi)^n\| + \|\sum_{n=0}^{N-1} \gamma^n (k')^\top (P^\pi)^n - \sum_{n=0}^{N-1} \gamma^n (k)^\top (P^\pi)^n\|$$

$$\leq O(\frac{\|k\|\gamma^N}{1 - \gamma}) + \|\sum_{n=0}^{N-1} \gamma^n (k)^\top (P^\pi)^n - \sum_{n=0}^{N-1} \gamma^n (k')^\top (P^\pi)^n\|$$

$$\leq O(\frac{\|k\|\gamma^N}{1 - \gamma}) + \|k - k'\|\|\sum_{n=0}^{N-1} \gamma^n (P^\pi)^n - \sum_{n=0}^{N-1} \gamma^n (P^\pi)^n\|$$

$$\leq O(\frac{\|k\|\gamma^N}{1 - \gamma}) + O(\frac{\|k - k'\|}{1 - \gamma}).$$

577 $\qquad\qquad\qquad\qquad\qquad\qquad\qquad\qquad\qquad\qquad\qquad\qquad\qquad\qquad\qquad\qquad\qquad\qquad\square$

578 **Computing Q-value given value function.** Let $v$ be $\epsilon_1$ approximation of robust value function $v_{\mathcal{U}}^{\pi}$,
579 that is

$$\|v - v_{\mathcal{U}}^{\pi}\|_{\infty} \le \epsilon_1.$$

580 We want to compute Q-value using the relation:

$$Q_{\mathcal{U}}^{\pi}(s,a) = R_{\mathcal{U}}^{\pi}(s,a) + \sum_{s,a} \pi_s(a) P_{\mathcal{U}}^{\pi}(s'|s,a) v_{\mathcal{U}}^{\pi}(s')$$

$$= R_0(s,a) + \gamma \sum_{s'} P_0(s'|s,a) v_{\mathcal{U}}^{\pi}(s') - \Omega_{\mathcal{U}}(v_{\mathcal{U}}^{\pi}, \pi).$$

581 where $\Omega_{\mathcal{U}_p^{\mathrm{s}}}(v_{\mathcal{U}_p}^{\pi}, \pi) = \frac{\pi_s(a)^{q-1}}{\|\pi_s\|_q^{q-1}} \left( \alpha_s + \gamma \beta_s \kappa_q(v_{\mathcal{U}_p}^{\pi\mathrm{s}}) \right)$ and $\Omega_{\mathcal{U}_p^{\mathrm{sa}}}(v_{\mathcal{U}_p}^{\pi\mathrm{sa}}, \pi) = \alpha_{sa} + \gamma \beta_{sa} \kappa_q(v_{\mathcal{U}_p}^{\pi\mathrm{sa}})$. Let
582 $Q$ be approximated from $v$ as

$$Q(s,a) = R_0(s,a) + \gamma \sum_{s'} P_0(s'|s,a) v(s') - \Omega_{\mathcal{U}}(v, \pi).$$

583 So we have,

$$\|Q_{\mathcal{U}}^{\pi}(s,a) - Q(s,a)\|_{\infty} = \gamma \|\sum_{s'} P_0(s'|s,a)(v(s') - v_{\mathcal{U}}^{\pi})\| + \|\Omega_{\mathcal{U}}(v,\pi) - \Omega_{\mathcal{U}}(v_{\mathcal{U}}^{\pi},\pi)\|$$

$$\le \gamma \epsilon_1 + \|\Omega_{\mathcal{U}}(v,\pi) - \Omega_{\mathcal{U}}(v_{\mathcal{U}}^{\pi},\pi)\|$$

$$\le \gamma \epsilon_1 + \|\beta\|_{\infty} \|\kappa_q(v) - \kappa_q(v_{\mathcal{U}}^{\pi})\|$$

$$\le \gamma \epsilon_1 + \|\beta\|_{\infty} S^{\frac{1}{q}} \epsilon_1, \qquad \text{(using lemma F.2)}$$

$$= O(S^{\frac{1}{q}} \epsilon_1).$$

584 This implies, $\|Q - Q_{\mathcal{U}}^{\pi}\|_{\infty} \le O(S^{\frac{1}{q}} \epsilon_1)$.

585 **Lemma F.2.** $\kappa_p$ is Lipschitz function, precisely

$$\|\kappa_p(v_1) - \kappa_p(v_2)\| \le S^{\frac{1}{p}} \|v_1 - v_2\|_{\infty} \le S^{\frac{1}{p}} \|v_1 - v_2\|_{\infty}.$$

586 *Proof.* Let $w_i \in \arg\min_w \|v_i - w\mathbf{1}\|_p$, then we have

$$\|\kappa_p(v_1) - \kappa_p(v_2)\| = \kappa_p(v_1) - \kappa_p(v_2), \qquad \text{(WLOG, assuming } \kappa_p(v_1) \ge \kappa_p(v_2))$$

$$= \min_w \|v_1 - w\mathbf{1}\|_p - \|v_2 - w_2\mathbf{1}\|_p, \qquad \text{(From definition)}$$

$$\le \|v_1 - w_2\mathbf{1}\|_p - \|v_2 - w_2\mathbf{1}\|_p, \qquad \text{(From definition of min operator)}$$

$$\le \|(v_1 - w_2\mathbf{1}) - (v_2 - w_2\mathbf{1})\|_p, \qquad \text{(Reverse triangle inequality)}$$

$$= \|v_1 - v_2\|_p$$

$$= [\sum_s (v_1(s) - v_2(s))^p]^{\frac{1}{p}} \le S^{\frac{1}{p}} \|v_1 - v_2\|_{\infty}.$$

587 $\qquad\qquad\qquad\qquad\qquad\qquad\qquad\qquad\qquad\qquad\qquad\qquad\qquad\qquad\qquad\qquad\qquad\qquad\qquad\qquad\qquad\qquad\square$

588 **Lemma F.3.** $Q_{\mathcal{U}_p^{\mathrm{sa}}}^{\pi}$ can be approximated to $\epsilon$ tolerance with the same complexity as complexity of
589 computing $v_{\mathcal{U}_p^{\mathrm{sa}}}^{\pi}$ to $S^{-\frac{1}{q}} \epsilon$.

590 *Proof.* Compute value function with $S^{-\frac{1}{q}} \epsilon$ tolerance. The rest of operations are insignificant. Rest
591 follows from the above. $\qquad\qquad\qquad\qquad\qquad\qquad\qquad\qquad\qquad\qquad\qquad\qquad\qquad\qquad\qquad\qquad\qquad\square$

592 ## F.2 Computing the Policy gradient.

593 Let $O_p^{\mathrm{sa}}(\epsilon)$ be the complexity to compute robust value function $v_{\mathcal{U}_p^{\mathrm{sa}}}^{\pi}$, upto $\epsilon$ tolerance, see [12] for
594 details. Calculate Q-value up to $\epsilon_1$ tolerance which requires $O_p^{\mathrm{sa}}(S^{-\frac{1}{q}} \epsilon_1)$ from lemma F.3. Let
595 $d_1$ and $d_2$ be $\epsilon_2$ approximation of $d_{P_0,\mu}^{\pi}$ and $d_{P_0,k}^{\pi}$ respectively, which is insignificant compared to
596 $O_p^{\mathrm{sa}}(S^{-\frac{1}{q}} \epsilon_2)$. Now let's approximate the gradient with $d_1, d_2, Q, \nabla\pi$ as in Theorem 5.1, which has a
597 complexity of $O(SA)$. Since the uncertainty set $\mathcal{U}$ is compact, all the quantities are bounded. And
598 there are $O(SA)$ operations in the Theorem 5.1, so taking $\epsilon_1, \epsilon_2 = O(\frac{\epsilon}{SA})$, we will get the $O(\epsilon)$ of
599 the gradient. Hence, the total complexity is $O_p^{\mathrm{sa}}(S^{-\frac{1}{q}-1} A^{-1} \epsilon)$ which is $\tilde{O}(S^2 A \log(\epsilon^{-1}))$, by hiding
600 log factors, see [12]. A similar analysis follows for the $s$-rectangular case.

# G Generalization to arbitrary norms

Here we focus on the generalization of our result to a general norm from the existing $\ell_p$ norm. We do it case by case.

## G.1 sa-rectangular robust MDPs.

Lets consider sa-rectangular uncertainty set $\mathcal{U} = \mathcal{U}^{\text{sa}}_{\|\cdot\|}$ constrained by $\|\cdot\|$ norm. Precisely, defined as

$$\mathcal{U}^{\text{sa}}_{\|\cdot\|} = (P_0 + \mathcal{P}) \times (R_0 + \mathcal{R}), \qquad \text{where} \qquad (\mathcal{P}, \mathcal{R}) = (\times_{s,a} \mathcal{P}_{sa}, \times_{s,a} \mathcal{P}_{sa}),$$

$$\mathcal{R}_{(s,a)} = \left\{ r \in \mathbb{R} \mid \|r\| \leq \alpha_{s,a} \right\}, \quad \text{and} \quad \mathcal{P}_{(s,a)} = \left\{ p \in \mathbb{R}^{\mathcal{S}} \mid \langle p, \mathbf{1} \rangle_{\mathcal{S}} = 0, \|p\| \leq \beta_{s,a} \right\}.$$

The robust Bellman operator $\mathcal{T}^{\pi}_{\mathcal{U}}$ can be evaluated as

$$(\mathcal{T}^{\pi}_{\mathcal{U}} v)(s) = \sum_a \pi_s(a) \left[ R(s,a) - \gamma \beta_{s,a} \kappa_{\|\cdot\|}(v) + \gamma \sum_{s'} P(s'|s,a) v(s') \right],$$

where variance function is defined as

$$\kappa_{\|\cdot\|}(v) := \min_{\langle u, \mathbf{1} \rangle_{\mathcal{S}} = 0, \|u\| \leq 1} \langle u, v^{\pi}_{\mathcal{U}} \rangle.$$

This can be used to compute the robust value function. Then the worst values can found using robust Bellman operator $\mathcal{T}^{\pi}_{\mathcal{U}}$ and robust value function $v^{\pi}_{\mathcal{U}}$ as

$$(P^{\pi}_{\mathcal{U}}, R^{\pi}_{\mathcal{U}}) \in \arg\min_{(P,R) \in \mathcal{U}} \mathcal{T}^{\pi}_{(P,R)} v^{\pi}_{\mathcal{U}}, \qquad [30].$$

It is easy to see that the worst values are given as

$$R^{\pi}_{\mathcal{U}}(s,a) = R_0(s,a) - \alpha_{s,a} \quad \text{and} \quad P^{\pi}_{\mathcal{U}}(\cdot|s,a) = P^{\pi}_0(\cdot|s,a) - \beta_{s,a} u^{\pi}_{\mathcal{U}},$$

where normalized-balanced value function $u^{\pi}_{\mathcal{U}}$ is a solution to

$$\min_{\langle u, \mathbf{1} \rangle_{\mathcal{S}} = 0, \|u\| \leq 1} \langle u, v^{\pi}_{\mathcal{U}} \rangle.$$

Observe that the worst kernel is still a rank-one perturbation of the nominal kernel. Hence, the robust occupation measure can be obtained using the Lemma 4.4 as

$$d^{\pi}_{\mathcal{U},\mu} = d^{\pi}_{P_0,\mu} - \gamma \frac{\langle d^{\pi}_{P_0,\mu}, \beta^{\pi} \rangle_{\mathcal{S}}}{1 + \gamma \langle d^{\pi}_{P_0, u^{\pi}_{\mathcal{U}}}, \beta^{\pi} \rangle_{\mathcal{S}}} d^{\pi}_{P_0, u^{\pi}_{\mathcal{U}}}, \tag{12}$$

where $\beta^{\pi}(s) = \sum_a \pi_s(a) \beta_{s,a}$. The last ingredient to compute RPG is robust Q-value which can be computed using robust value function and worst values. However, it can be computed directly using the following iterates:

$$Q_{n+1}(s) = \min_{(P,R) \in \mathcal{U}} \left[ R(s,a) + \gamma \sum_{s'} P(s'|s,a) v(s') \right]$$

$$= R(s,a) - \alpha_{s,a} - \gamma \beta_{s,a} \kappa_{\|\cdot\|}(v) + \gamma \sum_{s'} P(s'|s,a) v(s'),$$

as $Q_n$ converges to robust Q-value $Q^{\pi}_{\mathcal{U}}$ linearly.

The proofs of the above claims are easy or similar to the $\ell_p$ counterparts. Finally, computation of the variance function $\kappa_{\|\cdot\|}$ and normalized-value function $u^{\pi}_{\mathcal{U}}$ can be done via numerical convex optimization methods for general norms. However for the $\ell_p$ case, they can be obtained in concrete forms, hence we choose it for the main presentation.

## G.2 s-rectangular Case

Generalization of our methods to $s$-rectangular balls of a general norm is not straightforward and may not be possible for all kinds of norms. The crucial property of $\ell_p$ norm that we exploited to prove rank-one perturbation is 'decoupling', that is, for $x \in \mathbf{R}^{\mathcal{A} \times \mathcal{S}}$,

$$\|x\|^y_p = \sum_{a \in \mathcal{A}} \|x_a\|^z_w,$$

for some $w, y, z$. This holds for the $\ell_p$ norm with $w = y = z = p$. We leave the further analysis of this setting for future work.

## G.3 Generalization to non-norms

Further, generalization of our results to distance such as KL, can be tricky. The ability of our methods to compute RPG (particularly