# OpenReview forum: "Policy Gradient for Rectangular Robust Markov Decision Processes"
_NeurIPS.cc/2023/Conference — NeurIPS 2023 poster_

### Official Review · Reviewer_bn4R · 2023-07-03

**Soundness:** 3 good
**Presentation:** 4 excellent
**Contribution:** 3 good
**Rating:** 5
**Confidence:** 4

**Summary:**

The paper investigates policy gradient methods in the context of unknown transition kernels in Markov Decision Processes (MDPs). The authors focus on a robust approach, where the uncertain transition kernel is assumed to lie within a specific uncertainty set, known as a rectangular uncertainty set. The authors address the computation of robust policy gradients in this setting, aiming to achieve the same time complexity as in the non-robust case.

**Strengths:**

The combination of policy gradients with robust MDPs is an interesting and practical approach with promising initial results. The paper is well-written and accessible, making it relatively easy to understand.

**Weaknesses:**

There are a couple of issues regarding the modelling of the uncertainty set:
1) In Section 3.2.1 it is written that the radius $\beta$ of the uncertainty set should be “small enough” such that transition kernels of the form $P_0 + \mathcal{P}$ are well defined. This, however, requires the nominal transitions kernel $P_0$ to be strictly positive everywhere. Moreover, small positive values of $P_0$ are also not enough. Therefore, this modelling seems highly restrictive - particularly for high-dimensional settings.
2) The motivation for using rectangular ambiguity sets is not clear. In practice, ambiguity sets are often non-rectangular and derived from principles like maximum likelihood estimation. Could the method handle more general uncertainty sets?

**Questions:**

1) See the two questions raised as weaknesses?
2) Uncertainty sets are often chosen with the goal to achieve some statistical properties of the learned solution (here policy gradient). Is there a statistical motivation for choosing the specific uncertainty set you choose? Or in other words, why not choosing another uncertainty set than the one you chose?
3) In Section 3.1, what is the relation between $k$ and $\mu$?
4) How do you choose the radius $\beta$?
5) What is the motivation/justification for $\mathcal{U}$ being convex?
6) Line 106 has a typo. Should be $(P,R)$

**Limitations:**

1) Uncertainty set modelling is a) not well motivated and b) it seems rather restrictive. Please see the questions and points raised above.

---

> ### Author Rebuttal · Authors · 2023-08-10
>
> >*"In Section 3.2.1 it is written that the radius of the uncertainty set should be “small enough” such that transition kernels of the form are well defined. This, however, requires the nominal transitions kernel to be strictly positive everywhere...this modelling seems highly restrictive"*
>
> We apologize for the confusion this sentence has caused. We also agree that $P_0$ may have so small values that the uncertainty set would most likely include absurd MDP models. However, our results do not require all $P_0+P$-s in the uncertainty set to be well-defined.
>
> Like any existing work on robust MDPs [10, 17, 28], we treat an optimization problem under uncertainty set constraints. This approach does not question or rely on the realizability of all MDPs in the uncertainty set. Although all $P_0 + P$ should ideally be well-defined to obtain possibly less conservative behavior and more realistic uncertainty, robust RL methods do not rely on this assumption. Instead, they address the optimization problem regardless of whether or not all kernels in the uncertainty constraints are meaningful.
>
> Moreover, uncertainty sets are generally constructed based on statistical confidence intervals [28]. Although these may include absurd MDP models and lead to unecessarily conservative behavior, they concurrently hold generalization guarantees [29].
>
> >*"The motivation for using rectangular ambiguity sets is not clear. In practice, ambiguity sets are often non-rectangular and derived from principles like maximum likelihood estimation. Could the method handle more general uncertainty sets?"*
>
> We agree that ambiguity sets are non-rectangular in practice. In fact, this assumption is required to compute a worst model within polynomial time [28]. Solutions that relax this condition have been proposed in [I, II, III], but each one relies on a specific structure of the coupling. On the other hand, the $s$-rectangularity assumption is well-established in the robust RL literature [1, 3, 7, 8, 24]. Although relaxing it while preserving tractability would lead to more realistic solutions, it would deserve a full study on its own that goes beyond the scope of our work. Still, the construction of ambiguity sets from maximum likelihood estimation is compatible with rectangularity, and is actually described in [28, Sec. 5.2], [17, Sec. 5] and [10, Sec. 4].
>
> >*"In Section 3.1, what is the relation between $k$ and $u$."*
>
> The variable $k\in\mathbb{R}^{\mathcal{S}}$ is just used as a general notation in the definition of state visitation $d_{P,k}^{\pi}$. In our specific framework, $k$ is either the initial state distribution $\mu$ (first term in Eq. (3)), or the normalized robust value $u_{\mathcal{U}}^{\pi}$, i.e.,
> $d^\pi_\{P,u^\pi_{\mathcal{U}}}=(u^\pi_{\mathcal{U}})^\top(\mathbf{I}_\{\mathcal{S}}-\gamma P^\pi)^\{-1}$ (second term in Eq. (3)).
>
> >*"How do you choose the radius?"*
>
> The radius choice goes alongside the uncertainty set construction. As such, it is a challenging question that deserves its own analysis. Intuitively, the smaller the radius value, the less robust its optimal policy. On the other hand, if the radius is too large, the learned policy can be overly conservative. Therefore, determining the size of the uncertainty set leads to a robustness-conservativeness dilemma. Solutions that tighten the radius have been proposed in [IV]. Other approaches that update the uncertainty set online and adapt to incoming data can also be found in [V, VI]. However, these works tackle the conservativeness of robust policies, while we address the tractability of robust learning.
>
> >*"What is the motivation/justification for $\mathcal{U}$ being convex?"*
>
> The uncertainty set needs to be convex for computational reasons. Without convexity, the inner minimization problem (robust policy evaluation) or alternatively, the computation of the worst case model may be NP-hard (see [28, Table 1] and Table 1 in our work). The convex property is not hard to satisfy in practice, as uncertainty sets are generally constructed based on statistical confidence intervals or divergence constraints such as KL.
>
> >*"Line 106 has a typo. Should be $(P, R)$"*
>
> Right. We will fix this together with the two previous equations where we notice the same typo.
>
>
>
> **Additional references:**
>
> [I] Mannor, Shie, Ofir Mebel, and Huan Xu. "Robust MDPs with k-rectangular uncertainty." Mathematics of Operations Research 41.4 (2016): 1484-1509.
>
> [II] Mannor, Shie, Ofir Mebel, and Huan Xu. "Lightning Does Not Strike Twice: Robust MDPs with Coupled Uncertainty." ICML (2012).
>
> [III] Goyal, Vineet, and Julien Grand-Clement. "Robust Markov decision process: Beyond rectangularity." arXiv preprint arXiv:1811.00215 (2018).
>
> [IV] Petrik, Marek, and Reazul Hasan Russel. "Beyond confidence regions: Tight Bayesian ambiguity sets for robust MDPs." Advances in neural information processing systems 32 (2019).
>
> [V] Lim, Shiau Hong, Huan Xu, and Shie Mannor. "Reinforcement learning in robust Markov decision processes." Advances in Neural Information Processing Systems 26 (2013).
>
> [VI] Derman, Esther, et al. "A Bayesian approach to robust reinforcement learning." Uncertainty in Artificial Intelligence. PMLR, 2020.

---

> > ### Comment · Reviewer_bn4R · 2023-08-15
> > **After reading the rebuttal**
> >
> > I would like to thank the authors to answer my questions. Regarding the first question raised, I don't understand why you would want to consider uncertainty sets that contain "absurd" MDP models - this should lead to unnecessary conservatism. Related is the discussion about the choice of the radius - which also remains open (at least from the theoretical side). I do understand, however, that this questions may be beyond the scope of this work.

---

> > > ### Author Response · Authors · 2023-08-16
> > >
> > > We would not "want to", on the contrary. As mentioned in our response, "all $P_0+P$ should ideally be well-defined to obtain possibly less conservative behavior and more realistic uncertainty". On the other hand, the RMDP setting is model-based because the uncertainty set is given as input to the learning algorithm. Thus, the problem of interest is to solve an optimization problem, not to determine its constraints. In other words, except when inferred from data as in [I, II], the uncertainty set does not need to be estimated. In our case, the radius is not chosen but given. As you correctly mentioned, choosing the radius is beyond the scope of this work.
> > >
> > > Actually, even when it is given and its size is known, the uncertainty set is generally not intersected with the simplex, for simplicity. For example, in [III, Sec. 3], the authors occult the simplex constraints by focusing on a "proxy region".
> > >
> > > **Additional references:**
> > >
> > > [I] Lim, Shiau Hong, Huan Xu, and Shie Mannor. "Reinforcement learning in robust Markov decision processes." Advances in Neural Information Processing Systems 26 (2013).
> > >
> > > [II] Derman, Esther, et al. "A Bayesian approach to robust reinforcement learning." Uncertainty in Artificial Intelligence. PMLR, 2020.
> > >
> > > [III] Roy, Aurko, Huan Xu, and Sebastian Pokutta. "Reinforcement learning under model mismatch." Advances in neural information processing systems 30 (2017).

---

### Official Review · Reviewer_c24r · 2023-07-04

**Soundness:** 3 good
**Presentation:** 3 good
**Contribution:** 3 good
**Rating:** 7
**Confidence:** 4

**Summary:**

This paper proposes the robust policy gradient (RPG) algorithm as an efficient approach to solve rectangular RMDPs. The results are based on an explicit characterization of $\ell_p$-ball-constrained RMDPs, where the worst-case kernel can be viewed as a rank-one perturbation of the nominal kernel and the worst-case occupation frequency can also be written in closed form, so that the well-known policy gradient theorem can still be applied to calculate the sub-gradient for the RMDP. It is further pointed out in the appendix that these results are all closely related to a fundamental characterization of a constrained inner product minimization problem. The paper also redefines the robust value functions using the worst-case kernel, and shows its equivalence with the common definition in literature, so that it comes up with a more efficient way to calculate the value function of RMDPs, which constitutes as the last piece for evaluating the sub-gradient. Numerical simulations are presented to show the efficiency of the proposed algorithm.

**Strengths:**

1. The flow of this paper is very good. It provides the reader with enough background information to understand the framework, and the final algorithm naturally builds upon the building blocks established in the first few sections so that the readers can easily acquire the intuition.
2. The paper consists of a thorough literature review that justifies the place of this result in the spectrum of related works. The reviewer particularly appreciates the table presented in Section 2 that compares the contributions a full range of related works.
3. The contribution of this work is significant as viewed by the reviewer, in that it is among the first to apply policy gradient methods to solve RMDPs. The algorithm is presented with theoretical complexity guarantees, and the paper also discusses more efficient LP-based algorithms  in the $\ell_1$-constrained special case, which adds to the applicability of the algorithm.
4. The mathematical proofs seem correct to the reviewer in the form they are presented in the paper. (Note: since the proof cites quite a few theorems from e.g. [12], the reviewer took those published results for granted and didn't bother to check them.)

**Weaknesses:**

1. The numerical experiments seem relatively crude and incomplete, and the results are not convincing as viewed by the reviewer. The authors do not release the code for the simulations, and nor do they describe the settings in detail so that the results can be replicated. The presentation of results only focuses on running time but says nothing about the policy found by the algorithm, nor are they compared to any SOTA baselines. Experiments seem to be repeated with different random seeds, yet the deviation is only roughly reported in the appendix. Finally, figures cannot be found anywhere! The reviewer will be more positive if the results are more concrete, visualized, and reproducible.
2. This paper cites a lot of results from previous work like [12]. Readers unfamiliar with it (like me) may sometimes find it very uneasy to handle these cited results which basically scatter across the whole paper. It is advisable to have an independent preliminaries section that explains what is done in [12] and why we need them here.

**Questions:**

1. The reviewer urges the authors to design better experiments to show both the running time and the performance of the proposed RPG algorithm against a few other baseline algorithms. Results should ideally be presented in the form of figures.
2. In the introduction, [14] seems to be closely relevant to this paper, yet it is not discussed in details in the related works section, and nor does it appear in the table. How does this work compare with [14]? And how about the efficiency of the algorithm proposed in [14]?

**Limitations:**

The paper discusses about its limitations and future work in a satisfactory way.

---

> ### Author Rebuttal · Authors · 2023-08-10
>
> > *"numerical experiments seem relatively crude and incomplete"*
>
> For each experiment, we fixed a state-space size and an action space size. An environment was generated by readily sampling a transition kernel and a reward matrix with proper dimensions. This was taken as an input to RPG or LP-RPG. A table with all parameters, including those that do not appear in the appendix, can be found below:
>
> | Parameter  | Value |
> | ----------- | ----------- |
> | discount factor $\gamma$ | 0.9        |
> | initial distribution $\mu$ | uniform      |
> | $\epsilon$ tolerance error for policy evaluation      | 1e-4    |
> | learning rate for RPG update | 1e-3        |
> | number of RPG iterations | 3000        |
> | $\alpha$-radius (constant for all states/state-action pairs)  | 0.1        |
> | $\beta$-radius (constant for all states/state-action pairs)  | 0.01  $\cdot(\lvert\mathcal{S}\rvert\lvert\mathcal{A}\rvert)^{-1}$ |
>
> As mentioned in the appendix, the std was <10% for all the experiments, and in most cases between 1 and 2%. We avoided writing the exact values to make the result tables more readable. However, following your suggestion, we will display the exact std for each of our experiments in Tabs. 3 and 4.
>
> We additionally ran RPG ($s$ and $(s,a)$-rectangular cases), together with a classical PG baseline. The convergence plot can be found in the following link:
> https://www.dropbox.com/scl/fi/w0ob97xgejwxon6ccozbq/rpg_methods_s50a20seed50.png?rlkey=eb4dqawr38k2oobsjlppl2712&dl=0
>
> The green plot labeled 'nr' corresponds to the non-robust PG; the blue line is RPG on $s$-rectangular balls for the $\ell_2$-norm, whereas the orange plot represents its $(s,a)$-rectangular version.
>
> As we can see in the figure, the number of iterations required for RPG to converge is similar to that of PG. The performance of RPG upon convergence is slightly lower than PG, because the robust approach induces some conservativeness. In that respect, the performance in the $(s,a)$-case is slightly lower than the $s$-case, which confirms the additional conservativeness of $(s,a)$-rectangularity.
>
> We will add RPG's pseudo-code in the final version and release the full code upon publication.
>
>
>
> > *"This paper cites a lot of results from previous work like [12]. Readers unfamiliar with it (like me) may sometimes find it very uneasy to handle these cited results which basically scatter across the whole paper. It is advisable to have an independent preliminaries section that explains what is done in [12] and why we need them here."*
>
> We apologize for the lack of comprehensiveness and provide further explanation below.
>
> The work [12] refines the results of [3]. In both works, the motivation is to reduce the computational cost of robust dynamic programming. They do so using regularization, a well-known technique of robustification in supervised learning. More precisely, [3] establishes an equivalence between robust Bellman updates and regularized ones with a properly defined regularization function.
>
> Yet, when it comes to normed-ball ambiguity sets, the authors in [3] occult the simplex constraints in the transition uncertainty. This may lead to unneeded conservativeness: if not intersected with the simplex, ball-constraints alone may include elements that are not transition kernels while the learned policy accounts for them in the inner minimization. [12] fills this gap, so the value regularizer $\lVert v\rVert$ of [3] becomes $\kappa_q(v)$ in [12].
>
> These results are needed in the policy gradient derivation. In order to avoid solving an optimization problem at each gradient step of RPG (as proposed in [26]), we leverage the equivalence between robust RL and regularized RL established in [3, 12] to speed-up RPG updates.
>
> We will include these details in the final version of this work.
>
>
> > *"In the introduction, [14] seems to be closely relevant to this paper, yet it is not discussed in details in the related works section, and nor does it appear in the table. How does this work compare with [14]? And how about the efficiency of the algorithm proposed in [14]?"*
>
> The work [14] provides a convergence proof of robust policy mirror descent, which is different from our contribution. Moreover, full access to the gradient is assumed in [14], whereas we provide an explicit expression of RPG that can be estimated from data. Also, Li et al. [14] focus on $(s,a)$-rectangular uncertainty sets, while we additionally study robust policy optimization in the $s$-rectangular case. In fact, their restriction to the $(s,a)$-case prevents us from transposing their analysis to our setting. This is due to the fact that the standard robust Bellman operator on $Q$-functions can no longer be applied on $s$-rectangular sets.
>
> As of [26], it proves convergence of RPG with a sample complexity of  $O(\frac{1}{\epsilon^4})$. This result holds for arbitrary uncertainty sets as long as there is an oracle access to robust policy gradient. Otherwise,  the method they propose to compute the gradient is much more expensive and not scalable.

---

> > ### Comment · Reviewer_c24r · 2023-08-14
> > **Thanks for the responses!**
> >
> > I appreciate the efforts made by the authors to settle my concerns. The experiment part definitely looks better now, and the supplementary materials are also useful. Based on these improvements, I will raise my score to 7.

---

### Official Review · Reviewer_rkBw · 2023-07-07

**Soundness:** 4 excellent
**Presentation:** 4 excellent
**Contribution:** 4 excellent
**Rating:** 6
**Confidence:** 3

**Summary:**

The paper presents a policy gradient method for solving rectangular-robust MDPs. They propose closed forms for the worst-case reward and transition models. They also present robust Q-value forms that can be efficiently updated by a proper Bellman operator. Furthermore, they show an expression for the robust policy gradient that matches the non-robust PG complexity, and is therefore very fast.

**Strengths:**

The paper targets a specific problem, and is very well-written. I applaud the authors for their valuable work, and enjoyed reading through the paper without any significant issues. The notation used for the theory is standard and appropriately compact, and the theory itself is clear and insightful. The computational efficiency obtained by the proposed method is both supported by theory, and also shown to be significant-enough in practice. This proves, once again, that a well-tailored algorithm can produce significant speed-ups over generic solvers (e.g., linear programming or convex optimizers).

**Weaknesses:**

While this is a minor point, I think the authors could slightly increase external references and elaborate more to help (an otherwise unfamiliar) reader follow through the content. For instance, (1) in Line 100, it would only take the authors a few words to reference this inequality as the Holder inequality, (2) provide a second common name for terms (such as the credal set for the probability simplex, or the visitation frequency for the occupation measure), (3) briefly derive simple results with the proposed notation (e.g., the gamma-contraction property in Line 118). These are only a few examples, and will mostly help the reader warm-up and get comfortable with the derivations before jumping into the core part of the paper.

Also, while I understand that this is out of the scope of the paper, but due to its importance I should mention it. The authors could possibly comment deeper about the resiliency of their proposed approach to stochastic approximation. While there is some limited discussion in the paper (e.g., Lines 218) about this point, I would like this point to be discussed in more length. If in future, this method is adapted to a more realistic or model-free setting, it would be valuable to have the authors comment on the statistical aspects of this approach.

**Questions:**

I would like the authors to address the points I raised in the Weaknesses section, even though they don't point to any fundamental short-comings in the paper.

**Limitations:**

The paper portrays a fair description of its limitations. This includes, but isn't limited to, (1) the second paragraph of discussion (Lines 304-311), (2) Lines 262-267, and (3) Lines 218-222.

---

> ### Author Rebuttal · Authors · 2023-08-10
>
> We thank the reviewer for the positive feedback.
>
> > *"slightly increase external references and elaborate more to help (an otherwise unfamiliar) reader follow through the content"*
>
> Thank you for the suggestion. We will go over the text thoroughly to elaborate more on the wording and the mathematical derivations.
>
> > *"comment deeper about the resiliency of [the] proposed approach to stochastic approximation"*
>
> We agree with the reviewer, as the main objective of our work is to eventually enable the applicability of robust RL to more realistic settings. As mentioned in lines 218-222, the remaining challenge in our approach is to estimate the occupation measure $d^\pi_\{P_0, u_\{\mathcal{U}}^\{\pi}}$. We may use importance sampling techniques, but as in any off-policy approach, we need to control variance and bias here. This is an interesting direction that we leave for future work. Other points worth being checked theoretically and empirically are the sensitivity of RPG to the learning rate and to the step size in the policy parameters.
>
> Following the reviewer's suggestion, we will include more of this discussion in the final version.

---

> > ### Comment · Reviewer_rkBw · 2023-08-16
> > **Thank you for your response**
> >
> > I thank the authors for their response, and will maintain my initial score as the authors addressed all my concerns.

---

### Official Review · Reviewer_qyVV · 2023-07-21

**Soundness:** 3 good
**Presentation:** 3 good
**Contribution:** 3 good
**Rating:** 6
**Confidence:** 3

**Summary:**

This paper develops policy gradients for robust MDPs, and proves several interesting properties of the worst-case kernel and occupancy measure along the way. The paper also shows that the robust Bellman operator is just the vanilla Bellman operator, with regularization of the vector's generalized variance function. The key result is a robust gradient expression that only depends on nominal samples.

**Strengths:**

1. Except the large amount of newly introduced notation, this paper seems polished and well written.
2. Good discussion on time complexity, which is often absent from many prior works but crucially important for robust MDPs.
3. The proposed method only requires samples from the nominal data distribution, and appears more computationally efficient than prior convex optimization approaches.


**Weaknesses:**

1. Notation is a bit hard to follow and is defined all over the place throughout section 3. It would be helpful to have a centralized place where all notation is defined.
2. There is no "convergence to local/global optima" result.
3. The experimental results only show running time, but does not evaluate the robust MDP performance of the learned policies. Do all methods end up learning the same robust optimal policies?

**Questions:**

1. Why is there a p-norm on a scalar in def. of R_{(s,a)}?
2. In Sec 3.2.1., what if the perturbation has a negative entry, which could possibly make P_0 + P have negative entries as well? This is not so unrealistic if P_0 has small entries somewhere.
3. Can this result generalize to convex uncertainty sets, or does it crucially rely on the uncertainty set being a lp ball?
4. Looking at Table 1, is the main benefit of this work a better time complexity, rather than the ability to learn better policies? Convex optimization approaches that guarantee global optimum seem to only be worse in terms of time complexity.



**Limitations:**

Please see questions/weaknesses.

---

> ### Author Rebuttal · Authors · 2023-08-10
>
> We first thank the reviewer for the constructive remarks.
>
> > *"Notation is a bit hard to follow and is defined all over the place throughout section 3. It would be helpful to have a centralized place where all notation is defined."*
>
> Thank you for the suggestion. Although we tried to centralize it as much as possible, we wanted to put notations into context, which may have turned out to be too spread throughout the text.
> Yet, we agree that some definitions could appear in the Notation paragraph rather than in the middle of Sec. 3. For example, the definition of the variance and the mean function is general enough to move it to the Notation section. Perhaps also the regularization function $\Omega_q$ defined in Prop. 3.1. We will do that in the final version and centralize notations more carefully.
>
> > *"There is no "convergence to local/global optima" result."*
>
> Robust policy gradient has recently been shown to converge to a global optimum in [26]. Global convergence of our RPG method thus follows. This is hinted at in lines 92-96 and lines 147-151. We will emphasize this more clearly in the final version.
>
> > *"The experimental results only show running time, but does not evaluate the robust MDP performance of the learned policies. Do all methos end up learning the same robust optimal policies?"*
>
> Yes, they do. For a given nominal MDP and uncertainty set, all baselines should end up learning the same robust optimal policy. Our RPG distinguishes itself from other methods by its computational efficiency (see also our last answer below).
>
> > *"Why is there a p-norm on a scalar in def. of R_{(s,a)}?"*
>
> The reviewer is correct. Although it appears for consistency, the p-norm notation is redundant there. We will remove it and write $|r| \leq \alpha_{s,a}$ instead.
>
> > "*In Sec 3.2.1., what if the perturbation has a negative entry, which could possibly make P_0 + P have negative entries as well?*"
>
> Again, the reviewer is correct: even when all entries of $P_0$ are positive, they could be so small that $P_0 + P$ may have negative entries, and the uncertainty set possibly includes absurd MDP models.
>
> In fact, as with many existing works on robust MDPs [10, 17, 28], we tackle the constrained optimization problem without caring about the realizability of all constraints. Moreover, uncertainty sets are generally based on statistical confidence intervals [28]. Although these may include absurd MDP models and lead to unnecessarily conservative behavior, the robust approach holds generalization guarantees.
>
> In that respect, our sentence "the noise radius $\beta$ should be small enough so that transition kernels of the form $P_0 + P$ are well defined" (l. 129-130) is misleading. Although all $P_0 + P$ should ideally be well-defined to obtain possibly less conservative behavior and more realistic uncertainty, robust RL methods do not rely on this assumption. Instead, they address the optimization problem regardless of whether or not all kernels in the uncertainty constraints are meaningful.
>
> >*"Can this result generalize to convex uncertainty sets, or does it crucially rely on the uncertainty set being a lp ball?"*"
>
> Our result crucially relies on the 'rank-one-perturbation structure' of the worst transition kernel (Lemma 4.4). The $\ell_p$-ball uncertainty implies such property, which we establish in Thm. 4.6. However, whether any convex uncertainty set leads to the worst transition kernel being a rank-one perturbation of the nominal remains an open question. In fact, it would be interesting to investigate the structural properties needed on the uncertainty set for the rank-one perturbation to hold. We thank the reviewer for this remark.
>
> >"*Looking at Table 1, is the main benefit of this work a better time complexity, rather than the ability to learn better policies? Convex optimization approaches that guarantee global optimum seem to only be worse in terms of time complexity.*"
>
> The reviewer is correct: this work's main benefit is providing a substantially better time complexity than existing RPG methods. Like previous studies on robust policy gradient, our algorithm converges to an optimal robust policy (global optimum, see [26]). However, it does so within a much lower time complexity. This point is critical, as most robust RL methods cannot scale to deep domains because of their computational burden, whereas they provably ensure more stable behavior.

---

> > ### Comment · Reviewer_qyVV · 2023-08-17
> >
> > Thanks for your response. I keep my score.

---

### Official Review · Reviewer_2g5m · 2023-08-01

**Soundness:** 3 good
**Presentation:** 4 excellent
**Contribution:** 3 good
**Rating:** 6
**Confidence:** 3

**Summary:**

This present studied robust policy gradient to solve rectangular robust Markov decision process, where the authors provide a closed-form expression for the worst case of reward model and transition kernel. The provided explicit form of the robust policy gradient can be estimated from data with the same time complexity as its non-robust equivalent.

**Strengths:**

- Instead of previously performing max-min optimization approaches, this paper presents a closed-form expression for the worst case kernel and reward estimation; thus we can achieve similar time complexity.

- Empirical experiments demonstrate that the new proposed method can achieve similar complexity as non-robust settings, make the algorithm more practical than previous ones.

- The main part of the paper is easy to follow and understand the main idea.

**Weaknesses:**

- The techniques mainly replies on the properties of the normed-ball structure of the uncertainty set, I am not sure how practical it is in realistic settings such as continuous policy optimization?

- it is still unclear for real-world scenarios, how the method or the idea can be leveraged to derive a computational efficient and practical algorithm. To me, it seems that to get the optimal reward and kernel, it is quite difficult to estimate from empirical samples (continuous case).

**Questions:**

- Can authors give me a concrete example or explanation of the difference of the risk-averse approach and robust RL? the uncertainty set you mentioned?

---

> ### Author Rebuttal · Authors · 2023-08-10
>
> > *"Leveraging the method to derive a computationally efficient and practical algorithm in continuous policy optimization"*
>
> We thank the reviewer for the constructive remark.
>
> In fact, one of the goals in this work is to develop building blocks for robust continuous control. The derivation of an explicit robust policy-gradient is a first step towards this goal. On the one hand, our RPG provably trains a robust optimal policy (as opposed to the reward-robust model in [3] or $r$-contaminated kernel-robust model in [27]). On the other hand, it avoids the expensive convex optimization solvers required to find the worst MDP model [26]. Yet, we agree that some challenges remain in our RPG formulation, in particular how to estimate the occupancy measure $d^\pi_\{P_0, u^\pi_\{\mathcal{U}}}$ (see l. 218-222). Importance sampling may be used here, but we leave this for future work.
>
> As a final note, previous works that proposed to train a robust policy in a model-free setting either restricted to discrete control [15, I] or focused on a specific type of robustness criterion [3, 4, 27]. We  believe that devising an efficient RPG method with theoretical grounding paves the path towards more applications to robust continuous control.
>
>
> > *"A concrete example or explanation of the difference of the risk-averse approach and robust RL?"*
>
> We believe the reviewer refers to lines 72-75, where we briefly explain the difference between the two settings: the risk-averse approach focuses on the **aleatoric uncertainty**, i.e., the uncertainty internal to the system's stochasticity. Whereas the robust setting tackles the **epistemic uncertainty** due to ambiguity in the model. We further clarify this below.
>
> Risk-averse RL aims to overcome the **variability** of the return. For example, a mean-variance objective would aim to maximize return under variance constraints, or dually, minimize variance under minimal performance constraints. The moments considered there are taken with respect to one environment dynamics and reward model. General risk measures similarly formulate various probabilistic constraints under one fixed distribution.
>
> Differently, robust RL tackles **adversarial** outcomes in the return. Thus, a robust agent would maximize return under the worst environment dynamics. This is where the uncertainty set plays a role, as it defines over what set of environments to judge which is worst: if it is reduced to a singleton, we would recover the standard RL objective; whereas if it is a set of properly perturbed versions of the true environment, we would recover a risk-averse objective [23]. In the robust MDP case, the uncertainty set can be arbitrary provided that it is convex and rectangular.
>
> To give further intuition, consider a fully deterministic MDP, i.e., with deterministic transition and reward functions. In this setting, a risk-averse objective would necessarily reduce to a standard expected utility, whereas through its uncertainty set, the robust objective can address changes in the dynamics and/or the reward function (both being possibly stochastic). Although compatible (one may combine risk averseness with robustness in the objective function), the two approaches are different.
>
> [I] Derman, Esther, et al. "Twice Regularized Markov Decision Processes: The Equivalence between Robustness and Regularization." arXiv preprint arXiv:2303.06654 (2023).

---

> > ### Comment · Reviewer_2g5m · 2023-08-20
> > **Response**
> >
> > I would like to thank the authors for answering my questions. I will keep my score and vote for acceptance.

---

### Decision · Program_Chairs · 2023-09-21

**Decision:**

Accept (poster)

**Comment:**

This paper studies policy learning for robust MDPs with rectangular uncertainty sets (which allow the solution to be characterized by a robust Bellman equation) and develops a policy gradient approach. The paper is well executed, clear, interesting, and addressing a specific problem, as recognized by the reviewers. There was some hesitance on the significance with only weak recommendations but there was uniform agreement the paper is valuable. I concur with that consensus.